# Beta-Titanium Alloy Covered by Ferroelectric Coating–Physicochemical Properties and Human Osteoblast-Like Cell Response

**Marta Vandrovcova** [1], **Zdenek Tolde** [2], **Premysl Vanek** [3], **Vaclav Nehasil** [4], **Martina Doubková** [1], **Martina Trávníčková** [1], **Jan Drahokoupil** [3], **Elena Buixaderas** [3], **Fedir Borodavka** [3], **Jaroslava Novakova** [4] **and Lucie Bacakova** [1,*]

1    Laboratory of Biomaterials and Tissue Engineering, Institute of Physiology of the Czech Academy of Sciences, Videnska 1083, 14220 Prague, Czech Republic; marta.vandrovcova@seznam.cz (M.V.); Martina.Doubkova@fgu.cas.cz (M.D.); Martina.Travnickova@fgu.cas.cz (M.T.)
2    Department of Materials Engineering, Faculty of Mechanical Engineering, Czech Technical University Karlovo Sq. 13, 12135 Prague, Czech Republic; Zdenek.Tolde@fs.cvut.cz
3    Institute of Physics of the Czech Academy of Sciences, Na Slovance 2, 18221 Prague, Czech Republic; vanek@fzu.cz (P.V.); draho@fzu.cz (J.D.); buixader@fzu.cz (E.B.); borodav@fzu.cz (F.B.)
4    Department of Surface and Plasma Science, Faculty of Mathematics and Physics, Charles University, V Holesovickach 747/2, 18000 Prague, Czech Republic; nehasil@mbox.troja.mff.cuni.cz (V.N.); jaroslava.novakova@mff.cuni.cz (J.N.)
*    Correspondence: Lucie.Bacakova@fgu.cas.cz; Tel.: +42-02-9644-3743

**Abstract:** Beta-titanium alloys are promising materials for bone implants due to their advantageous mechanical properties. For enhancing the interaction of bone cells with this perspective material, we developed a ferroelectric barium titanate ($BaTiO_3$) coating on a Ti39Nb alloy by hydrothermal synthesis. This coating was analyzed by scanning electron and Raman microscopy, X-ray diffraction, piezoresponse force microscopy, X-ray photoelectron spectroscopy, nanoindentation, and roughness measurement. Leaching experiments in a saline solution revealed that Ba is released from the coating. A progressive decrease of Ba concentration in the material was also found after 1, 3, and 7 days of cultivation of human osteoblast-like Saos-2 cells. On day 1, the Saos-2 cells adhered on the $BaTiO_3$ film in higher initial numbers than on the bare alloy, but they were less spread, and their initial proliferation rate was slower. These cells also contained a lower amount of $beta_1$-integrins and vinculin, i.e., molecules involved in cell adhesion, and produced a lower amount of collagen I. This cell behavior was attributed to a higher surface roughness of $BaTiO_3$ film rather than to its potential cytotoxicity, because the cell viability on this film was very high, reaching almost 99%. The amount of alkaline phosphatase, an enzyme involved in bone matrix mineralization, was similar in cells on the $BaTiO_3$-coated and uncoated alloy, and on day 7, the cells on $BaTiO_3$ film attained a higher final cell population density. These results indicate that after some improvements, particularly in its roughness and stability, the hydrothermal ferroelectric $BaTiO_3$ film could be promising coating for improved osseointegration of bone implants.

**Keywords:** metallic bone implants; electroactive coating; electrical charge; polarization; ferroelectricity; cell adhesion; cell proliferation; osteogenic differentiation; bone tissue engineering

## 1. Introduction

Bone implants are of increasing importance in orthopedic, dental, plastic, and re-constructive surgery for treating or replacing hard tissues damaged by various diseases (osteoarthrosis, osteoarthritis, tumors, inborn malformations) or by trauma (e.g., traffic accidents and industrial accidents, sports injuries etc.). Bone implants can generally be divided into two groups: temporary implants, and long-term/permanent implants. Temporary implants, e.g., splints, external fixators, wires, screws, or spikes, are intended to be removed

after the bone damage has healed. These implants need to be bioinert, i.e., they should not support the adhesion and growth of cells of the bone, so that the implant can be removed easily [1]. Permanent implants include replacements of small and large joints (e.g., hip, knee, shoulder, and trapeziometacarpal joints) and stomatological implants. These types of implants, especially the parts of them that are integrated into the bone (e.g., the stems and cups of hip joint replacements), need to support the adhesion, growth, and phenotypical maturation of osteoblasts, including the production of mineralized bone matrix by these cells, in order to achieve good osseointegration of the implant. This good osseointegration improves the secondary stability of the implant, prolongs its lifetime and, at the same time, decreases the likelihood of failure and the need for reoperation [2,3].

Nowadays, titanium-aluminum-vanadium (Ti6Al4V) alloy is frequently used for constructing clinically-used permanent bone implants because of its relatively low price, relatively good mechanical properties and good corrosion resistance [4,5]. However, due to a relatively high elastic modulus of this alloy (more than 100 GPa) in comparison with the native bone tissue (about 30 GPa in cortical bone), a stress-shielding effect appears, and this can cause bone resorption and loosening of the implant [6]. This problem can be mitigated by the use of alloys based on Ti with $\beta$-structure, i.e., a crystallographic phase with a body-centered cubic (BCC) lattice, which has a principally lower elastic modulus than the $\alpha-$ or $\alpha + \beta$ structure that is usually present in currently used titanium-based implants [7,8]. These materials include in particular a new titanium-niobium (TiNb) alloy (a binary system with a $\beta$-structure, i.e., $\beta$-TiNb), and a number of other Ti–Nb, titanium-niobium-tantalum (Ti–Nb–Ta) and titanium-zirconium-niobium (Ti–Zr–Nb) alloys. The elastic modulus of these materials decreases to ~80 GPa at ~25 wt.% of Nb for TiNb, to ~50 GPa at ~25 wt.% of Nb and 6.25% Zr for TiNbZr, and to ~75 GPa at ~23 wt.% of Ta and 10 wt.% of Zr for TiTaZr [9,10]. At the same time, these alloys in general have a higher corrosion resistance [11]. Moreover, these alloys can overcome another problem associated with the Ti6Al4V-based implants, namely the potential release of harmful elements V and Al, which can induce, among others, high levels of prostaglandin $E_2$ that plays an important role in regulation of the bone resorption process [12,13]. On the contrary, the Ti, Nb, Zr, and Ta elements have been reported as non-toxic [14], which paved the way for intensive development of Ti alloys with these elements as promising materials for surgical implants [6].

In addition to the novel types of alloys, a wide range of modifications of the physical and chemical properties of the material surface has been developed in order to enhance its attractiveness for the adhesion and growth of bone cells. These modifications include changes in chemical composition of the material surface (e.g., introduction of various functional groups, which can increase its polarity and wettability), optimization of the surface roughness and morphology [2,3], and particularly creation of electrically active surfaces [15–18].

An interesting type of electrically active surfaces are ferroelectric surfaces. Ferroelectric materials are smart electroactive materials capable of spontaneous polarization, i.e., of induction of electric dipole moment in their molecules and generation of electrical charge on their surface. The polarization of ferroelectric materials occurs through changes in the dimensions of their lattice. These changes can be induced by two main stimuli, i.e., temperature and mechanical stress; the ferroelectric materials are therefore both pyroelectric and piezoelectric [19].

Ferroelectric materials are of ceramic or polymeric nature. A well-known ferroelectric ceramic material is lead zirconate titanate (PZT), widely used for industrial applications, but it is highly cytotoxic and unsuitable for biomedical application, due to its content of Pb. The main lead-free ceramics more suitable for biomedical applications include e.g., barium titanate ($BaTiO_3$), lithium niobate ($LiNbO_3$), sodium-potassium, or potassium-sodium niobate (referred to as NKN or KNN, respectively), lithium tantalate ($LiTaO_3$), or magnesium silicate ($MgSiO_3$) [20–23]. Polymeric materials with ferroelectric properties widely tested for biomedical applications include polyvinylidene fluoride (PVDF) and

poly(L-lactic acid) (PLLA). Some mineral and polymeric components of the native bone tissue, namely hydroxyapatite and collagen, also show ferroelectric properties [22,23].

In most studies on ferroelectric ceramics, these materials have been used as free-standing materials, i.e., in the form of polycrystalline or single-crystalline plates [20] or porous ceramic scaffolds [24]. They have also been applied in the form of (nano)particles immersed in simulated body fluid [25], added into cell culture media [26] or injected into laboratory animals [27]. The particles have also been incorporated into synthetic and natural matrices, such as poly(lactic-co-glycolic acid) (PLGA) membranes [28], electrospun nanofibrous PLLA scaffolds [29], alginate scaffolds [30], hydrogels based on decellularized and demineralized bone matrices [31], or have been combined with piezoelectric PVDF [32]. However, direct deposition of ferroelectric films, which are by definition also piezoelectric, on metallic materials designated for bone implantation is relatively rare. In our earlier study, $BaTiO_3$ films were fabricated on titanium and TiNb substrates by pulsed-laser deposition (PLD) [16]. However, the disadvantages of PLD of $BaTiO_3$ on TiNb, performed at a high temperature (600–700 °C), and also of rapid thermal annealing crystallization, is that both of these methods are conditioned by the Pt interlayer between TiNb and $BaTiO_3$ [17]. This solution is not ideal, because Pt can act as cytotoxic and can induce inflammatory reactions [33,34]. Another possible method for growing $BaTiO_3$ films with ferroelectric properties is physical vapor deposition (PVD) [35]. The PVD method is also suitable for oxide preparation as a basis for hydrothermal synthesis of a $BaTiO_3$ film. Using this method, however, biocompatible ferroelectric and piezoelectric films have been prepared on single crystalline substrates. Ferroelectric $BaTiO_3$ films were obtained after post-annealing [35]. Other methods for growing a $BaTiO_3$ film with ferroelectric properties on titanium implants are electrostatic spray pyrolysis (ESP) [36], which combines the biocompatibility of a ceramic with the high mechanical strength of a metal, and also methods based on electrochemical principles, such as micro-arc oxidation [37]. Another promising method, which we intended to explore in this study, is the hydrothermal method. This method can be expected to produce stable $BaTiO_3$ film on TiNb substrates and to coat homogeneously complex shapes of various bone implants. Recently, this method was successfully applied to coat porous 3D-printed Ti6Al4V scaffolds with $BaTiO_3$ in order to improve their interaction with osteogenic cells in vitro and osseointegration in vivo [18].

This study is therefore focused on preparing a ferroelectric $BaTiO_3$ coating on a β-titanium alloy (Ti39Nb) by the hydrothermal method. This coating was then characterized in terms of its surface roughness and morphology, thickness, hardness, chemical composition, piezoelectric response, and also stability, evaluated by changes of Ba concentration in these coating after elution in a physiological saline solution. The effect of $BaTiO_3$ coating on the adhesion, spreading, viability and subsequent growth of cells was evaluated in cultures of human osteoblast-like Saos-2 cells on these materials. This cell line was chosen due to its resemblance with primary osteoblasts in terms of activity of alkaline phosphatase (ALP) and expression of osteocalcin, which are important markers of osteogenic cell differentiation [38]. The presence and amount of selected specific markers of cell adhesion, namely $\beta_1$-integrins and vinculin, and early markers of osteogenic cell differentiation, namely ALP and collagen I, were also evaluated in Saos-2 cells by measuring the intensity of fluorescence after immunofluorescence staining of these molecules.

## 2. Materials and Methods

### 2.1. Preparation of Substrates for Ferroelectric Coating

The β-titanium Ti39Nb alloy (i.e., a titanium alloy with 39 wt.% of niobium), which served as the substrate material for deposition of ferroelectric film, was manufactured in UJP Praha a. s. (Prague, Czech Republic). The alloy was prepared by a powder metallurgy technique from a mixture of 61 wt.% of titanium and 39 wt.% of niobium powder. Subsequently the alloy was rotary forged at 1050 °C to a rod shape with diameter of 10 mm, and was heat-treated by a solution annealing at 850 °C for 0.5 h. Samples with 2 mm thickness were manufactured from the final rod. They were ground to grit 4000, polished

by a colloidal silica suspension (0.05 µm) and finally washed by an ultrasonic cleaner in acetone [16].

### 2.2. Preparation of Ferroelectric BaTiO$_3$ Coating by Hydrothermal Synthesis

The hydrothermal coating of Ti39Nb substrates by a BaTiO$_3$ film was carried out by a procedure based in principle on the study by Zhu et al. [39]. First, the substrates were cleaned ultrasonically in acetone, etched in 6 M HCl (Lach-Ner, Neratovice, Czech Republic, G.R., 35%) for 20 min, rinsed by deionized water and dried. Then the substrates were annealed in flowing oxygen in a tube furnace at 500 °C for 24 h; the substrates were protected from contamination by a cleaned silica glass tube.

In the next step, 1.47 g BaCl$_2$.2H$_2$O (Lachema, Brno, Czech Republic, p.a.) was dissolved in 30 mL of reboiled deionized water. 1.20 g NaOH (Sigma-Aldrich, St. Louis, MO, USA, reagent grade, ≤1.0% NaCO$_3$) was added to the solution and dissolved, forming Ba(OH)$_2$ by the reaction. The substrates were placed in a lab-made Teflon holder, and then into a Teflon insert of the Berghof DAB-2 autoclave. The solution was filtered through a Whatman syringe single-use 0.45 µm PVDF filter directly to the autoclave insert in order to remove potential BaCO$_3$ precipitates. All these steps were performed under argon flow to reduce the formation of barium carbonate. The hydrothermal reaction of Ba(OH)$_2$ and TiO$_2$ was performed at 250 °C for 6 weeks. The coated substrates were washed in deionized water, soaked in 20 vol.% acetic acid (Lach-Ner, Neratovice, Czech Republic, G.R., 98%) to dissolve the possible residues of BaCO$_3$, finally washed by deionized water and dried. Three series of samples were prepared by the above-described method with the same parameters. The third series was used for X-ray diffraction (XRD) grazing incidence examination only.

### 2.3. Basic Characterization of the Substrate and Coating

The surface roughness of the Ti39Nb substrate and BaTiO$_3$ coating was measured with a Hommel T1000 Tester (JENOPTIK, Jena, Germany). The nano-hardness of the coating was measured near to the surface using the NanoTest system (Micro Materials Ltd., Wrexham, UK) with a Berkovich indenter. The applied load for the film was 3 mN, and the depth was 160 nm. From the measured values, the reduced modulus of elasticity was determined. The thickness of the BaTiO$_3$ film was evaluated by a calo-tester (CSM Instruments, Peuseux, Switzerland) and electron microscope. The surface morphology of the substrate and of the film was evaluated with a JEOL JSM 7600-F scanning electron microscope (SEM; JEOL, Tokyo, Japan) with energy-dispersive X-ray spectroscopy (EDS) analysis, using secondary electron detectors (SEI, LEI) and a backscattered electron detector (LABE). A JEOL 2200FS high-resolution transmission electron microscope (HRTEM; JEOL, Tokyo, Japan) was used to observe the high resolution section of the BaTiO$_3$ film.

### 2.4. X-ray Diffraction (XRD), Micro-Raman Spectroscopy, and Piezoresponse Force Microscopy (PFM)

The oxidized substrates and the hydrothermally deposited coating were characterized by XRD using an X'Pert Pro X-ray diffractometer (PANalytical, Almelo, Netherlands,) equipped with Co tube ($\lambda$ = 0.1789 nm); symmetrical scans with divergent beam were used for overall characterization. The grazing incident geometry with parallel beams geometry was used to enhance the signal from the layers closer to the material surface. The diffraction patterns were processed according to Rietveld using the TOPAS 3 software [40], and the original structural model was taken from the Inorganic Crystal Structure Database (ICSD).

In order to verify the ferroelectric (and therefore piezoelectric) character of BaTiO$_3$ at room temperature, the coating was characterized by micro-Raman spectroscopy (with a RM-1000 RENISHAW Raman Microscope, Renishaw, Wotton-under-Edge, UK). Raman spectra were collected in individual points of the film, and also along line scans, using a wavelength of 514.5 nm under vertically polarized light. The laser spot had a size of ~3 µm, and the resolution of the spectra was better than 2 cm$^{-1}$. For comparison, BaTiO$_3$ ceramics was measured by the Raman microscope as well.

Piezoresponse force microscopy (PFM) was used for further confirmation of the piezoelectric character of $BaTiO_3$ film. PFM measurements were performed by an Asylum Research (Oxford Instruments, Goleta, CA, USA) Cyphes S atomic force microscope (AFM) system in a Vector PFM mode. An Adama diamond cone probe FM-LC with a stiffness of 10 N/m was used. The frequency and amplitude of the alternating voltage, applied to the tip, was set to 740 kHz and 1 V for vertical PFM and to 1250 kHz and 1 V for lateral PFM.

### 2.5. X-ray Photoelectron Spectroscopy (XPS) and Stability of The Coating

Surface analysis of the prepared $BaTiO_3$ film was performed using XPS in an ultra-high vacuum chamber with a base pressure better than $2 \times 10^{-9}$ Torr. The spectra were obtained using an Omicron EA$-$125 electron energy analyser (Omicron, Taunusstein, Germany) and dual anode X-ray source. The aluminium $K\alpha_{1,2}$ line with a primary energy of 1486.6 eV was used to stimulate the emission of photoelectrons.

The $BaTiO_3$ film was studied as-prepared without any additional cleaning. After this measurement, it was immersed in a physiological saline solution (i.e., 0.9 mg of NaCl in 100 mL of distilled water) at 38 °C in order to study the chemical stability of the surface and potential release of Ba. The samples were exposed to the saline solution for 1 week, then for additional 3 weeks, and finally for additional 12 weeks (i.e., for 1, 4 and 16 weeks in total). After every particular step, the chemical composition of the sample surface, particularly the concentration of Ba, was determined by XPS. In addition to these long-time experiments, also short-time experiments were performed by the same manner but only for samples exposed to the saline solution for 1, 3 or 7 days. The reason was to compare the Ba concentration in these samples with values obtained on the corresponding surfaces after 1-, 3-, and 7-day long cultivation of Saos-2 cells (described below). Since the cells adhered on the surfaces exhibited a relatively complicated structure of the O and C characteristic peaks, it was not possible to distinguish between the contributions of the cells and that of the cultivation substrate to these peaks. This situation disabled the correct calculation of Ba and Ti concentrations. Therefore, the ratio of the measured intensities of peaks belonging to these elements was used to estimate the changes of Ba concentration in the samples and the release of Ba from the samples.

### 2.6. Cell Seeding

For cell cultivation, the samples of bare and $BaTiO_3$-coated Ti39Nb (thickness 2 mm, diameter 10 mm, area 0.79 $cm^2$) were disinfected with 70% ethanol for 2 h. Subsequently, they were washed with phosphate-buffered saline (PBS; Sigma-Aldrich, St. Louis, MO, USA, Cat. No. P4417) and were inserted into 24-well polystyrene cell culture plates (TPP, Trasadingen, Switzerland; well diameter 1.5 cm). The samples were then seeded with human osteoblast-like Saos-2 cells (European Collection of Cell Cultures, Salisbury, UK, Cat. No. 89050205). Each well contained 28,000 cells (i.e., about 15,000 cells·$cm^{-2}$) and 1.5 mL of McCoy's 5A medium (Sigma-Aldrich, St. Louis, MO, USA, Cat. No. M4892), supplemented with 15% fetal bovine serum (FBS; Sigma-Aldrich, St. Louis, MO, USA, Cat. No. F7524) and gentamicin (40 µg $mL^{-1}$, LEK, Ljubljana, Slovenia). Microscopic glass coverslips (Menzel Glaser, Braunschweig, Germany; diameter 12 mm, area 1.1304 $cm^2$) were used as reference samples. The cells on the samples were cultured for 1, 3, and 7 days at 37 °C in a humidified air atmosphere containing 5% $CO_2$. For each experimental group and time interval, three samples were used.

### 2.7. Evaluation of Cell Number and Cell Population Doubling Time

On day 1 after seeding, the cell number was determined on three samples for each experimental group. In addition, these samples were used for determining the cell viability and the size of the cell spreading area. The cell viability was determined on one sample per each experimental group by the LIVE/DEAD viability/cytotoxicity kit for mammalian cells (Invitrogen, ThermoFisher Scientific, Waltham, MA, USA, Cat. No. L3224) according to the manufacturer's protocol. Briefly, the samples were rinsed with PBS), and were incubated

for 5–10 min at room temperature in a mixture of two of the following probes: calcein AM, a marker of esterase activity in living cells, emitting green fluorescence, and ethidium homodimer-1, which penetrated into dead cells through their damaged cytoplasmic membrane and produced red fluorescence. The live and dead cells were then counted on microphotographs taken under an epifluorescence microscope (Olympus IX 51, DP 70 Digital Camera, Olympus, Tokyo, Japan, objective ×10). For each experimental group, 16 independent microscopic fields were taken.

The remaining two samples for each experimental group were rinsed with PBS, fixed with 70% frozen ethanol (room temperature, 20 min) and stained with a combination of two fluorescence dyes: Texas Red $C_2$-maleimide, which stains proteins of the cell membrane and cytoplasm (Invitrogen, ThermoFisher Scientific, Waltham, MA, USA, Cat. No. T6008; 20 ng·mL$^{-1}$ of PBS; red fluorescence) and Hoechst #33258, which stains the cell nuclei (Sigma-Aldrich, St. Louis, MO, USA, Cat. No. B1155; 5 µg mL$^{-1}$ of PBS; blue fluorescence). Both dyes were applied for 1 h at room temperature. The number of cells and their shape on the material surface were evaluated on microphotographs taken under an IX 51 microscope, equipped with a DP 70 Digital Camera (both from Olympus, Japan, objective ×10). For each experimental group, pictures of 36 independent microscopic fields (i.e., 18 fields from each sample) were taken. These pictures were also used for measurement of the size of cell spreading area (see below the Section 2.9.).

On day 3 after seeding, the cell number was evaluated on three samples for each experimental group, stained by immunofluorescence against $\beta_1$-integrins or vinculin, i.e., markers of cell adhesion. The cells were rinsed twice with PBS and fixed with 70% ethanol (room temperature, 20 min). The non-specific binding sites for antibodies were blocked with 1% bovine serum albumin in PBS containing 0.05% Triton X−100 (Sigma-Aldrich, St. Louis, MO, USA, Cat. No. X100) for 20 min at room temperature. The samples were then incubated with primary monoclonal antibodies, namely mouse anti-human $\beta_1$-integrin (MilliporeSigma, Burlington, MA, USA, Cat. No. MAB1981; one sample) or mouse anti-human vinculin (Sigma-Aldrich, St. Louis, MO, USA, Cat. No. V9131; one sample). Each antibody was diluted in PBS to concentration of 1:200 and applied overnight at 4 °C. The remaining one sample per each experimental group served as a negative control, where PBS was applied overnight instead of the primary antibody. After rinsing with PBS, the secondary antibody, represented by goat anti-mouse F(ab')2 fragment of IgG conjugated with Alexa Fluor® 488 (Invitrogen, ThermoFisher Scientific, Waltham, MA, USA, Cat. No. A11017; dilution 1:1000 in PBS; green fluorescence), was added to all samples for 1 h at room temperature. The cell nuclei were then counterstained with Hoechst #33258 (5 µg mL$^{-1}$ of PBS), and the filamentous actin (F-actin) was visualized by staining with phalloidin conjugated with TRITC (Sigma-Aldrich, St. Louis, MO, USA, Cat. No. P1951; dilution 1:100 in PBS; red fluorescence). Both dyes mixed in one solution were applied for 1 h at room temperature. Finally, the cells were rinsed twice in PBS photographed under an epifluorescence microscope (Olympus IX 51, DP 70 Digital Camera, Tokyo, Japan, objective ×10). For each experimental group, pictures of 48 independent microscopic fields (i.e., 16 from each sample) were taken. Representative images were also taken under Leica SPE confocal microscope (Leica Microsystems GmbH, Wetzlar, Germany).

On day 7 after seeding, the cell number was evaluated on three samples for each experimental group, stained by immunofluorescence against alkaline phosphatase (ALP) or against type I collagen, i.e., early markers of osteogenic cell differentiation. The method of immunofluorescence staining was applied by the same way as in the previous paragraph. The primary antibodies used on day 7 were: rabbit polyclonal anti-alkaline phosphatase (liver, bone, kidney; ThermoFisher Scientific, Waltham, MA, USA, Cat. No. PA5-21332, dilution 1:200; one sample) or rabbit polyclonal anti-type I collagen (CosmoBio Co., Ltd., Tokyo, Japan, Cat. No. LSL-LB-1197, dilution 1:200; one sample). The remaining one sample per each experimental group served as a negative control, where PBS was applied instead of the primary antibody. The secondary antibody was represented by goat anti-rabbit F(ab')2 fragment of IgG conjugated with Alexa Fluor® 488 (Invitrogen, ThermoFisher

Scientific, Waltham, MA, USA, Cat. No. A11070; dilution 1:1000; green fluorescence). The cell nuclei were counterstained with Hoechst #33258 (5 µg mL$^{-1}$ of PBS). Finally, the cells were photographed under an epifluorescence microscope (Olympus IX 51, DP 70 Digital Camera, Tokyo, Japan, objective ×20). For each experimental group, pictures of 48 independent fields (i.e., 16 from each sample) were taken.

The cell numbers obtained on days 1, 3, and 7 after seeding were also used for calculating the cell population doubling time (DT) between days 1 and 3, or between days 3 and 7, using the following equation:

$$DT = log2 \frac{t - t_0}{(logN_t - logN_{t_0})} \tag{1}$$

where $t_0$ and $t$ represent earlier and later time intervals after seeding, respectively, and $Nt_0$ and $Nt$ represent the number of cells at these intervals.

### 2.8. Evaluation of the Intensity of Fluorescence

The immunofluorescence staining was also used to evaluate the presence and amount of specific markers of adhesion and osteogenic differentiation in Saos-2 cells on the tested materials. The markers of cell adhesion, stained on day 3 after cell seeding, included integrin adhesion receptors with β$_1$-chain (β$_1$-integrins) and vinculin, an integrin-associated protein of focal adhesion plaques. The markers of the osteogenic differentiation, stained on day 7 after cell seeding, included ALP and type I collagen. The amount of these markers in cultures of Saos-2 cells was estimated by the intensity of green fluorescence. To measure this intensity, 16 microphotographs of randomly selected fields from each sample were taken under an epifluorescence microscope at the same exposure time for each specific marker. The intensity of fluorescence was then measured using a custom-made fluorescent image analyser software (Matejka R. ALICE: Fluorescent Image Analyser, version 1.0); for a detailed description of the image analysis, see [1]. The intensity of fluorescence was then normalized per cell. The fluorescence intensity of the negative control, i.e., samples stained only with a secondary antibody without primary antibodies, was subtracted.

### 2.9. Measurement of the Size of Cell Adhesion Area

Cells stained with Texas Red C$_2$-maleimide and Hoechst #33258 on day 1 after seeding were also used for measuring their spreading area on microphotographs. The size of the cell area projected on the material was measured using Atlas Software (Tescan, Brno, Czech Republic). Cells that developed intercellular contacts were excluded from the evaluation. For each experimental group, two independent samples (containing 708–2510 cells in total) were evaluated.

### 2.10. Statistical Analysis

The quantitative data was presented as mean ± Standard Deviation (S.D.) for physico-chemical analyses, or as mean ± Standard Error of Mean (S.E.M.) for biological analyses. The statistical analyses were performed using SigmaStat (Jandel Corporation, San Jose, CA, USA). The multiple comparison procedures were carried out by the ANOVA, Student-Newman-Keuls Method or Dunn's Method. The value $p \leq 0.05$ was considered significant.

## 3. Results and Discussion

### 3.1. Basic Properties of The Batio$_3$ Film and Ti39Nb Substrate

In this study, a ferroelectric BaTiO$_3$ film on Ti39Nb alloy substrate was successfully prepared by a hydrothermal synthesis. Basic properties of this film and the substrate, namely their roughness, thickness, modulus of elasticity and hardness, are given in Table 1. The thickness of the BaTiO$_3$ film is about 1 µm, which is big enough for the film to be considered as a bulk. It is apparent that the roughness of the BaTiO$_3$ film is markedly higher than that of the original Ti39Nb substrates. The roughness of this film is in the submicron-/micron-scale, while the roughness of the bare Ti39Nb substrates is in the

nanoscale. As a result, the hardness and the reduced modulus of elasticity of BaTiO$_3$ film showed a relatively high deviation variance, because the indentation was influenced by a disparity between the peaks and pits of crystals in the film.

**Table 1.** Values of the basic parameters of substrate [16] and of BaTiO$_3$ film.

| Film/Substrate | Roughness $R_a$ [µm] | Hardness [GPa] | Reduced Modulus of Elasticity $E_r$ [MPa] | Thickness [µm] |
|---|---|---|---|---|
| Film (BaTiO$_3$) | 0.93 ± 0.05 *** | 3.20 ± 1.30 | 142.00 ± 35.00 *** | 0.94 ± 0.06 |
| Substrate (Ti39Nb) | 0.02 ± 0.01 | 3.10 ± 0.10 | 95.00 ± 2.50 | N/A |

Mean ± S.D. from 3 regions (roughness), 16 regions (hardness and $E_r$) and 4 regions (thickness) measured on one sample for each experimental group. Statistical significance: *** $p \leq 0.001$ (ANOVA, Student-Newman-Keuls Method).

Scanning electron microscopy (SEM) and transmission electron microscopy (TEM) revealed that in comparison with the surface of the original Ti39Nb substrate, the surface of the BaTiO$_3$ film consisted of a system of crystal grains (Figure 1).

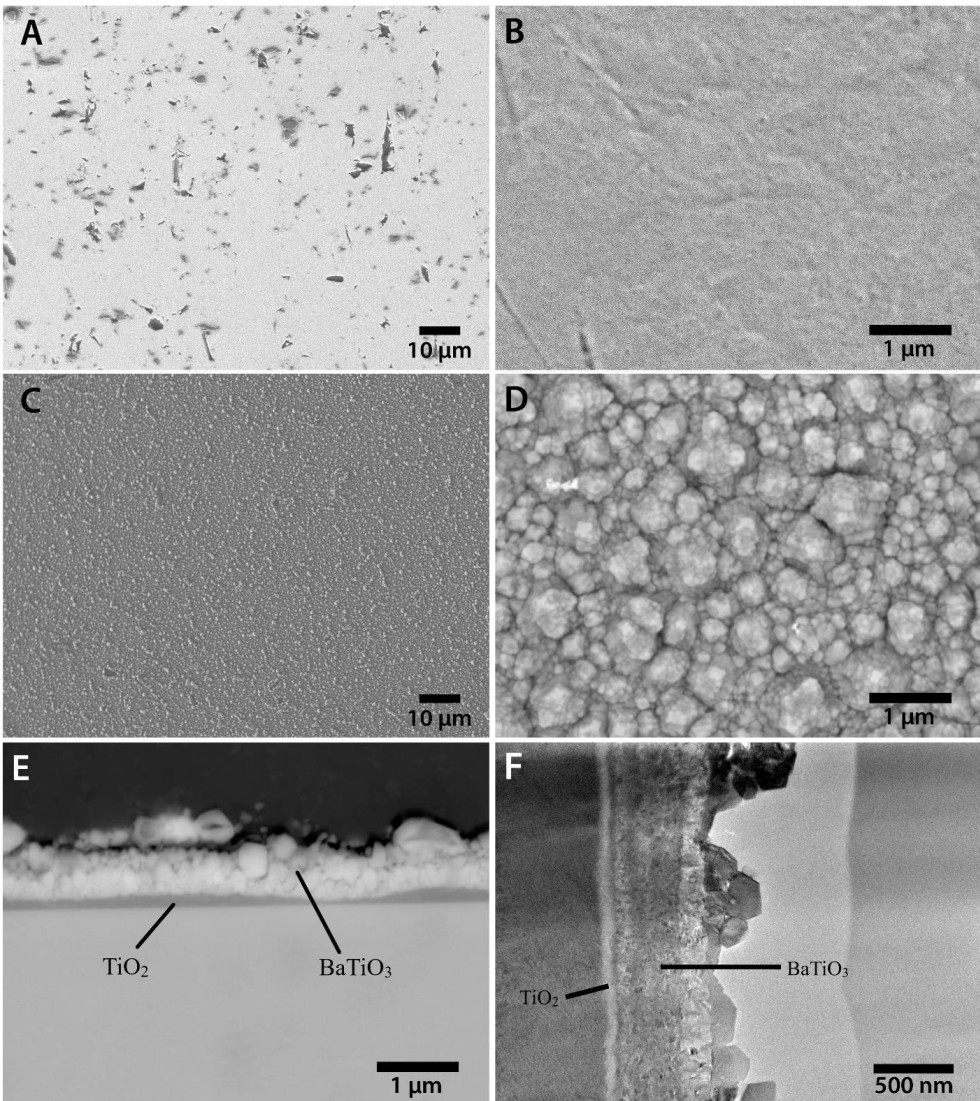

**Figure 1.** Representative SEM images of surfaces of the Ti39Nb substrate (**A**,**B**) and of the BaTiO$_3$ film (**C**,**D**). The images (**E**,**F**) show the cross-section of the BaTiO$_3$ film. The image (**E**) was obtained by SEM, the continuous BaTiO$_3$ film is visible and it is possible to identify individual crystals growing from the original TiO$_2$ layer. The image (**F**) was obtained by TEM and the cross-section of the BaTiO$_3$ film (rotated perpendicularly) is shown in high resolution. Magnification 1000× (**A**,**C**), 20,000× (**B**,**D**,**E**), 50,000× (**F**). Five samples of each material type were evaluated, and representative images were selected.

EDS (Energy-dispersive X-ray spectroscopy) analysis shows maps of chemical composition in section of the film. The aluminum layer on the film comes from an aluminum foil used as a protective coating preventing damage of the material during cutting and grinding (Figure 2).

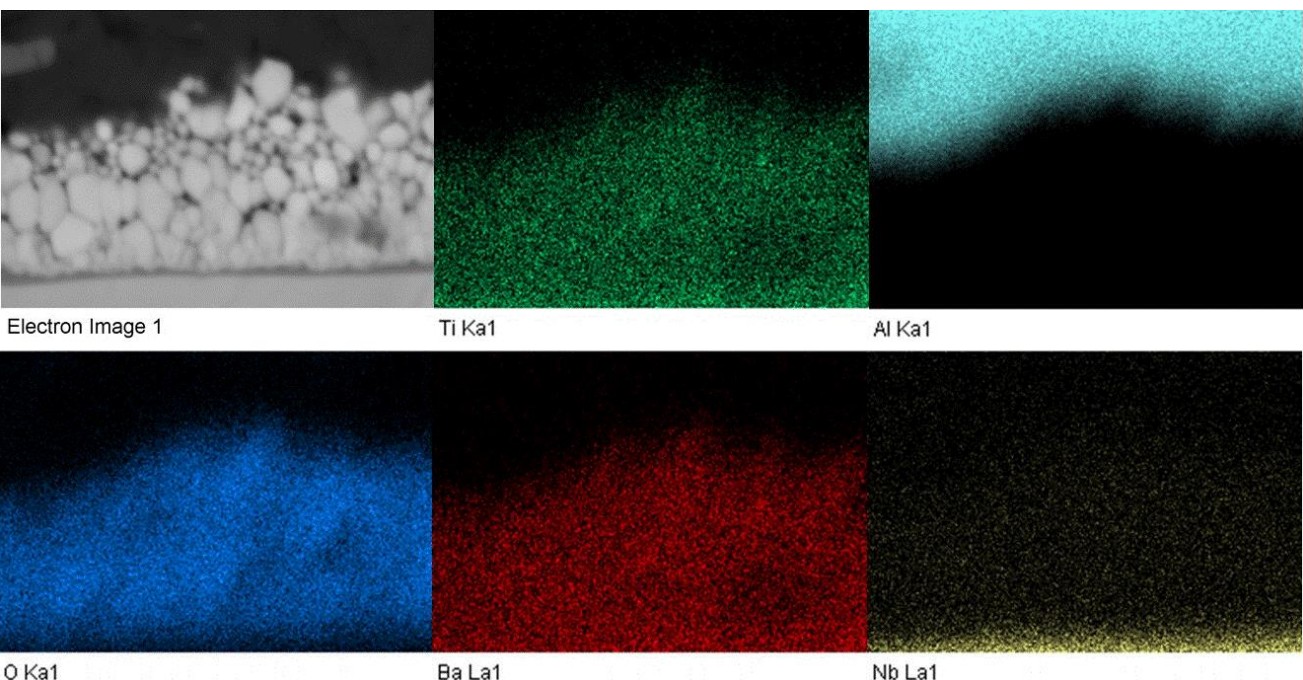

**Figure 2.** The energy-dispersive X-ray spectroscopy (EDS) analysis is showing the chemical composition throughout the $BaTiO_3$ film. The colors and their intensity indicate the presence (and amount) of each chemical species, which are designated under each picture (Ti, Al, O, Ba, Nb). The EDS analysis is a proof of the $BaTiO_3$ existence in the whole film; the K and L are indications of a shell to which the electron transitions during the bombardment. The analysis was performed in five regions of a $BaTiO_3$-coated sample, and representative images were selected.

### 3.2. XRD Analysis of Ti39Nb Substrates and BaTiO₃ Film

On the surface of oxidized Ti39Nb substrates, rutile $TiO_2$, and probably an $Im-3m$ cubic phase, were detected by XRD. The $Im-3m$ cubic phase has the same space group but a larger lattice parameter (3.35 Å) compared with the original Ti39Nb phase (3.28 Å). This increase of the lattice parameter can be explained by the presence of oxygen in the lattice and by a larger content of Nb compared to the initial material before oxidation, similar to the findings by Taylor et al. (1967), reporting the lattice parameter equal to 3.32 Å for $Nb_{0.94}$ $O_{0.060}$ [41]. Some Nb can also substitute the Ti in the rutile structure. No pure Nb-oxide phase was detected. The symmetrical scans for the original and oxidized Ti39Nb substrates, together with a grazing incident (0.5°) scan for the oxidized substrate, are plotted in Figure 3A. For the grazing incident measurement, no other peaks that corresponded to $TiO_2$ rutile phase were observed. For the symmetrical scan of oxidized Ti39Nb substrates, a strong peak at ca. 44.35° appeared. This peak probably corresponded to the original Ti39Nb phase (diffraction 110) with lower Ti and richer O concentrations. However, it was only a supposition, because the other characteristic peaks of the mentioned phase were not well distinguishable.

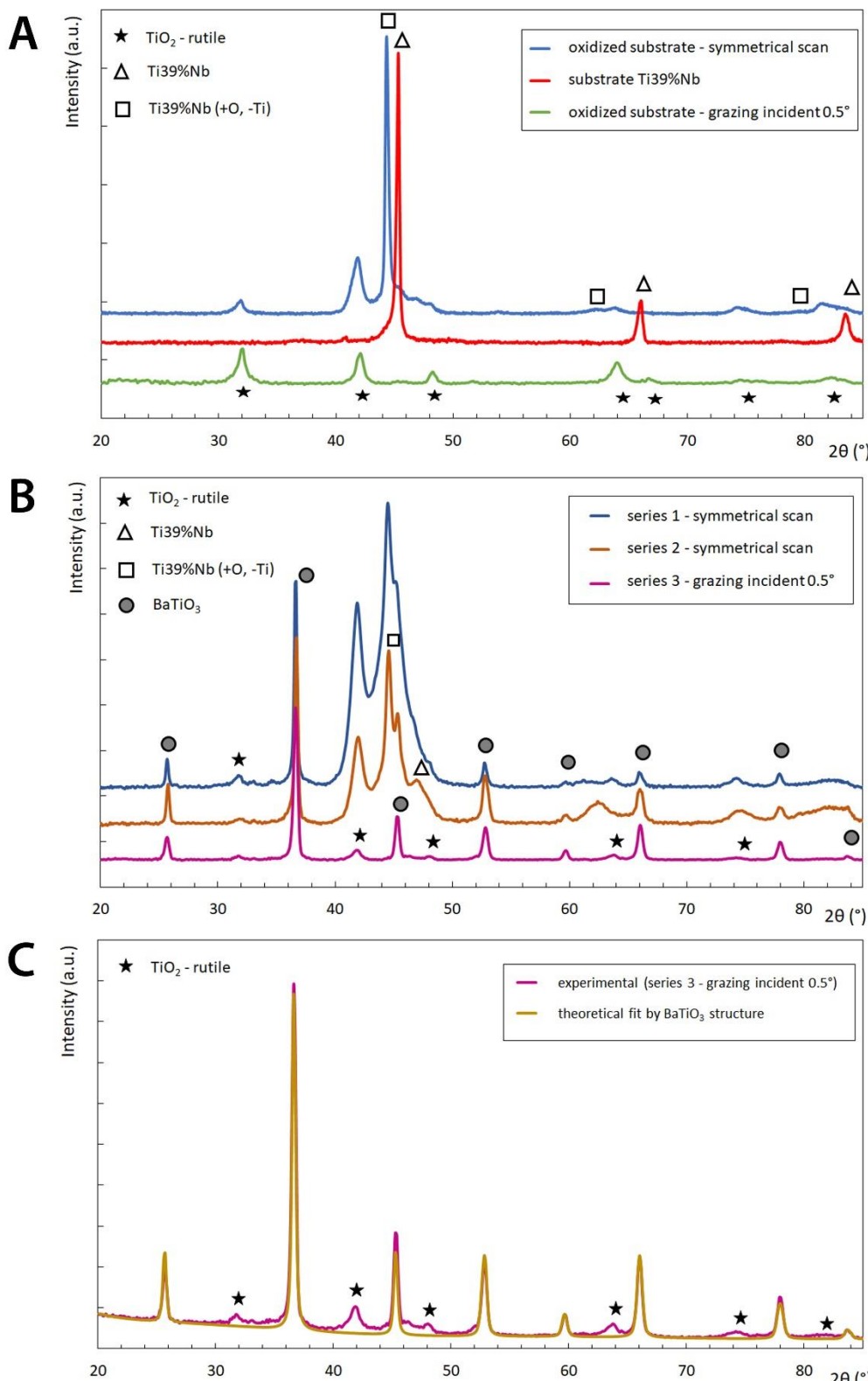

**Figure 3.** The diffraction patterns of the original and oxidized Ti39Nb substrates (**A**) and of BaTiO$_3$ film deposited on Ti39Nb substrates (**B**). XRD data for Series 3 sample measured with grazing incident angle of 0.5° fitted by BaTiO$_3$ phase. Not-fitted intensity corresponds to TiO$_2$ rutile phase (**C**). The analysis was performed on almost the entire surface (0.79 cm$^2$) of one sample per each experimental group (i.e., non-oxidized Ti39Nb, oxidized Ti39Nb, and BaTiO$_3$ of series 1, 2 and 3).

On hydrothermally-coated samples, the symmetrical scan detected nanocrystalline $BaTiO_3$ (crystallite size about 40 nm from the Rietveld refinement) together with about 35% of rutile $TiO_2$. The strong peak on ca. 44.3° from Ti39Nb substrate is also present. The presence of $TiO_2$ in the $BaTiO_3$/Ti39Nb samples implies that the hydrothermal reaction of $TiO_2$ with $Ba(OH)_2$ was not complete. The broad diffraction peaks of rutile $TiO_2$ and rutile $TiNbO_4$ overlap and it is not possible to distinguish them. The grazing incidence (0.5°) XRD with low penetration of X-rays detected mainly $BaTiO_3$ phase in the surface layers. The diffraction patterns are plotted in Figure 3B. The structure fit using $BaTiO_3$ phase on series 3 sample (measured with incident angle 0.5°) that proofs $BaTiO_3$ phase as a main phase in surface region is plotted in Figure 3C.

It is known that the dimensions of the crystallites (size effect) together with their chemical composition have a significant influence on the development of ferroelectricity on the material surface [42]. In our case, the size of the crystallites determined by Rietveld refinement of XRD was about 40 nm, which allows the development of stable spontaneous polarization, although ferroelectricity can be weaker than in bigger crystals. It is also obvious that most of the crystal grains seen by SEM (Figure 1D) are composed of several crystallites.

### 3.3. Raman Spectroscopy of Ti39Nb Substrates and BaTiO3 Film

Raman spectroscopy was chosen to confirm the structure of the film, as this technique is non-destructive, and easy to use at room temperature. In addition, this method is able to distinguish the specific phase of the material and its ferroelectric state. Micro-Raman spectroscopy detected the $BaTiO_3$ and $TiO_2$ rutile phases in the deposited film in accordance with XRD (Figure 4). No Nb-oxide was detected. Several features in the spectra (see Figure 4) prove the presence of $BaTiO_3$. In bulk form, this material is ferroelectric at room temperature [43]. This phase is well signalized by the tiny Raman peak near $300\ cm^{-1}$, which undoubtedly indicates the presence of the ferroelectric tetragonal phase of $BaTiO_3$ [44,45]. However, in comparison with $BaTiO_3$ ceramics, this peak is much less pronounced. It looks to be slightly more distinct in the sample from series 2 than the peak in the sample from series 1, although the difference is not significant. Therefore, the ferroelectricity of the deposited $BaTiO_3$ film seems to be much weaker than in $BaTiO_3$ ceramics, probably due to the so-called size effect [42], i.e., gradual loss of ferroelectricity at decreasing nanocrystallite size. The $520\ cm^{-1}$ band seen in $BaTiO_3$ ceramics is very weak and shifted to lower frequencies ($\sim490\ cm^{-1}$) in $BaTiO_3$/Ti39Nb coating; this phenomenon could be caused by a strain effect within the film. The presence of rutile peaks in the spectra is evident, as indicated by the strong peaks marked by arrows in Figure 4.

### 3.4. Piezoresponse Force Microscopy (PFM) of BaTiO3 Film

Piezoresponse force microscopy (PFM) technique was used in order to confirm the piezoelectric character of the $BaTiO_3$ film and to figure out the distribution of spontaneous polarization in particular grains of the film. Figure 5 shows a 5 μm × 5 μm topography image of the $BaTiO_3$ film surface and PFM images, namely a vertical amplitude and phase and a lateral amplitude and phase of a PFM signal. The topography image is in a good agreement with the SEM image (Figure 1D). By analyzing PFM images, we can conclude that the $BaTiO_3$ film has non-zero piezoelectric response and the spontaneous polarizations of individual grains are randomly oriented and do not have any preferred direction within the studied area.

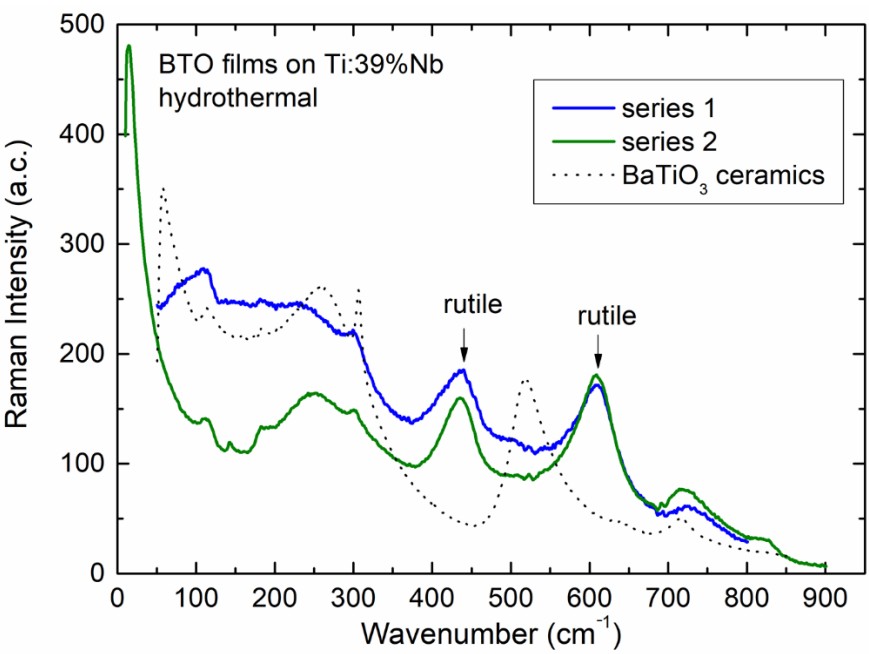

**Figure 4.** Raman spectra of $BaTiO_3$ film on Ti39Nb substrates at different points and from different batches (series 1 and series 2). For comparison, the spectrum of pure $BaTiO_3$ ceramics is shown. Arrows indicate the rutile peaks. The analysis was performed in five regions on each material sample, and a representative graph was selected.

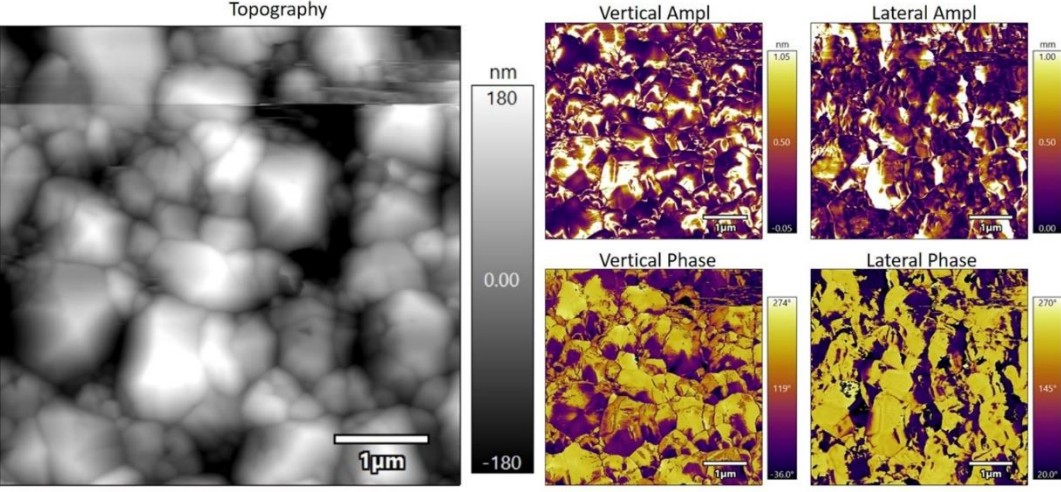

**Figure 5.** Topography and piezoresponse force microscopy (PFM) images of a $BaTiO_3$ film. The analysis was performed in a $5 \times 5$ $\mu m^2$ region on two $BaTiO_3$-coated samples, and representative images were selected.

### 3.5. XPS Analysis of BaTiO₃ Film on Ti39Nb Substrates

Wide photoelectron spectra of as-prepared samples and samples after treatment in the saline solution are presented in Figure 6. Principal peaks are described. Generally, Barium, Titanium, Oxygen, Niobium and Carbon peaks can be distinguished in the XPS spectra. It is clearly visible that after a longer time of treatment in the saline solution, the intensity of Ba 3d doublet decreases relatively to the signals of other elements, especially to Ti 2p and O 1 s.

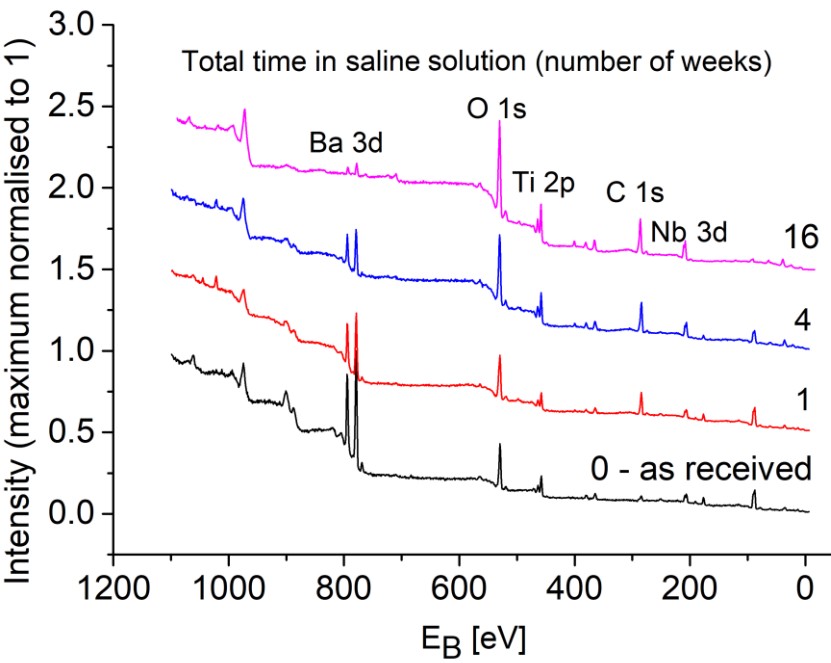

**Figure 6.** Wide photoelectron spectra of a BaTiO$_3$ film deposited on Ti39Nb. The sample was immersed in the physiological saline solution, and the dependence of Ba release from the sample on the total time of immersion is visible in the spectra. Particular total times of immersion (i.e., 1, 4, and 16 weeks) are specified near every spectrum. The analysis was performed on a circle of 5 mm in diameter on one sample per experimental group.

For quantitative analysis, intensities of particular peaks were used. Quantitative analysis was performed by a method of relative sensitivity factors. Shirley background was subtracted to obtain the proper area of peaks. To characterize BaTiO$_3$, we used only Ba, Ti and O-lattice peaks. Oxygen peak exhibits a complicated structure, and it can be fitted to several simple peaks appropriate to the particular chemical species. Lattice oxygen, which belongs to the BaTiO$_3$ compound and which is chemically bonded to Ba and Ti, can be observed on the binding energy about 529.6 eV [46–48]. Other chemical forms of oxygen can be interpreted as adsorbed oxygen, water molecules, and together with carbon, as the carbon-like impurities. The total Nb concentration, concluded from the quantitative analysis, is low, i.e., about 3% of Nb occurs in the sample surface as a residual contribution from the base TiNb material, and it is in the form of oxide. We do not suppose that it influences the total chemical composition and properties of the samples significantly, and we therefore do not include it into the following discussion.

Thus, taking in the account only Ba, Ti and O-lattice, the relative concentrations of particular elements were calculated and presented in Table 2. Values of the concentration ratios of Ba/Ti and O/(Ba + Ti) were added. It is evident that the as-prepared sample exhibits almost the stoichiometric ratio of Ba:Ti:O concentrations in BaTiO$_3$, i.e., 1:1:3, respectively. The Ba/Ti and O/(Ba + Ti) ratios are close to 1 and 1.5, respectively. Considering a typical error of the quantitative analysis performed by XPS, which can reach about 20% of the calculated value, we can conclude that the chemical composition of the prepared samples is satisfactory.

**Table 2.** Relative concentrations of elements contained in BaTiO$_3$ films, determined by XPS, and Ba/Ti and O/(Ba + Ti) concentration ratios.

| Procedure | Concentration [%] | | | | |
|---|---|---|---|---|---|
| | Ba | Ti | O | Ba/Ti | O/(Ba + Ti) |
| As prepared | 17.5 | 19.6 | 62.9 | 0.89 | 1.69 |
| Plus 1 week in saline sol. | 10.5 | 17.9 | 71.6 | 0.59 | 2.52 |
| Plus 3 weeks in saline sol. | 4.9 | 20.0 | 75.1 | 0.24 | 3.02 |
| Plus 12 weeks in saline sol. | 1.1 | 23.8 | 75.2 | 0.05 | 3.02 |

After treatment in the saline solution, we observed a decrease of barium concentration in the BaTiO$_3$ film deposited on Ti39Nb samples, which is clearly apparent in Table 2. This release of barium is the most visible after the first week, later it slows down, but after 16 weeks of treatment, the surface of the prepared film contained almost no barium. Using XPS, we confirmed the release of barium into the saline solution by detecting traces of Ba in the NaCl obtained after drying the used saline solution (data not presented here). A similar phenomenon of the dissolution of barium ions from BaTiO$_3$ nanoparticles in aqueous suspensions was described in detail in an earlier study performed by Tripathy et al. [49]. Thus, it should be admitted that if the BaTiO$_3$ film developed in our study is used as coating for bone implants, barium would be released into the human body. The amount of Ba found by XPS in the dried NaCl from the used saline solution was therefore recalculated for a film of a 100 cm$^2$ area, which is an implant area possibly coated with BaTiO$_3$. From Na and Cl intensities we have roughly estimated that the amount of released barium in this case is much lower than the toxic intake of this element that is about 200 mg [50]. Moreover, the growth and other performance of cells on the barium titanate surface was very satisfactory, enabling the coverage of the implant surface with cells within a relatively short period of several days (see below).

Short-term leaching experiments of samples with BaTiO$_3$ surfaces exposed to the saline solution for 1, 3, or 7 days also revealed a decrease of Ba concentration in these surfaces, similarly as in the BaTiO$_3$ surfaces after 1-, 3-, or 7-day long cell cultivation (Figure 7). As explained in Section 2.5, we used the ratio of Ba and Ti intensities to compare the decrease of barium in both types of surfaces. The decrease in Ba/Ti ratio in surfaces after one day of cell cultivation was similar as the value obtained after one day of leaching in the saline solution. However, after 3 and 7 days of cell cultivation, the loss of Ba from BaTiO$_3$ surfaces covered with cells was lower than from the cell-free surfaces exposed for 3 or 7 days to the saline solution. This finding suggests a protective effect of the cell coverage on the Ba release from BaTiO$_3$ film. In addition, proteins adsorbed on the BaTiO$_3$ surfaces from the serum of the culture medium, together with extracellular matrix proteins deposited by the cells during cultivation, could also mitigate the release of Ba from the material. Last but not least, the kinetics of Ba release may differ in the saline solution and in the cultivation medium, which is a richer solution containing a wide spectrum of ions, glucose, amino acids, vitamins and proteins. Analogically, the behavior of calcium also differed in various environments. When calcium phosphate ceramics was exposed to distilled and deionized water, calcium was released from the material, while in cell culture media, calcium was captured by the material and depleted from the media [51].

Contrary to barium, which was considerably decreasing in the samples during their elution in the saline solution, titanium conserves its concentration in the range of error (Table 2). A small increase after 16 weeks of leaching is probably a result of the barium decrease, when the titanium concentration relatively increased. The development of the oxygen concentration is similar.

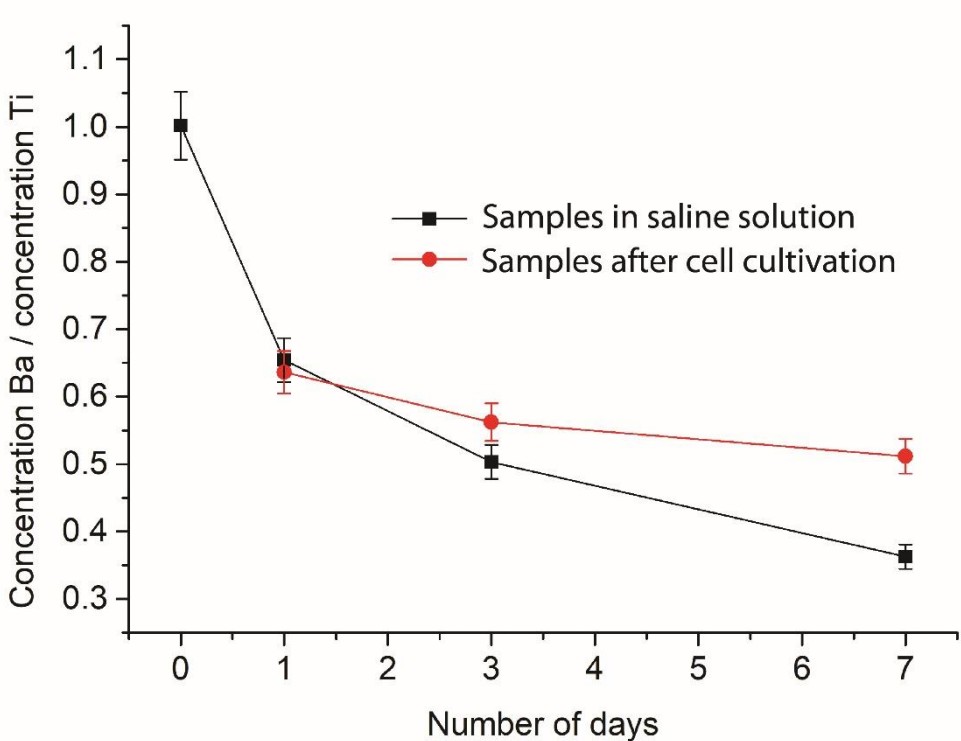

**Figure 7.** Decrease of the Ba/Ti concentration ratio in BaTiO$_3$-coated Ti39Nb samples after 1, 3 and 7 days of leaching in the physiological saline solution (black) or after 1, 3, or 7 days of cell cultivation (red). Error bars mean 5% of values presented in the graph, which is usually supposed inaccuracy in XPS results. The analysis was performed on a circle of 5 mm in diameter on one sample per each experimental group and time interval.

### 3.6. Attachment, Spreading, and Proliferation of Cells on the Materials

On days 1 after seeding, the numbers of initially adhered Saos-2 cells on Ti39Nb coated with BaTiO$_3$ were significantly higher than on the bare alloy (Figure 8A). This may be explained by the ferroelectricity of the BaTiO$_3$ film. It is believed that on the electrically-active materials, (i.e., electrically-charged or electroconductive materials), the cell adhesion-mediating proteins, present in the serum supplement of the culture medium (e.g., vitronectin, fibronectin), are adsorbed in an advantageous geometrical conformation, which facilitated the binding of cells through their adhesion receptors, mainly integrins [52]. Another explanation is a higher surface roughness of BaTiO$_3$ film in comparison with the bare Ti39Nb alloy. Due to the irregularities (i.e., depressions and prominences) on their surface, these films may provide a larger area for attachment for a larger number of cells than flatter surfaces, namely Ti39Nb and reference microscopic glass coverslips. However, a higher material surface roughness is advantageous only for the initial cell attachment, when the seeded cells, suspended in the culture medium, contact the material in their round shape. When the cells start to spread on the material surface, i.e., to increase their area contacting the material, the irregularities on the material surface may hamper this process. The cells usually try to spread over tens of micrometers, and irregularities in micrometer scale, present on the BaTiO$_3$ film may be limiting for this process [2,52,53]. As a result, the cells on the BaTiO$_3$-coated Ti39Nb looked smaller, were more rounded and their spreading area, i.e., their area projected on the material, was significantly smaller than on the bare alloy (Figure 9A,B). In addition, on the Ti39Nb alloy, characterized by a nanoscale surface roughness, the cell spreading is significantly better than on completely flat microscopic glass coverslips. It is believed that the nanoscale surface roughness resembles the nanoarchitecture of the natural extracellular matrix and is beneficial for the adhesion and subsequent growth of cells, particularly osteoblasts, due to preferential adsorption of vitronectin from the serum of the culture medium on nanostructured surfaces [52].

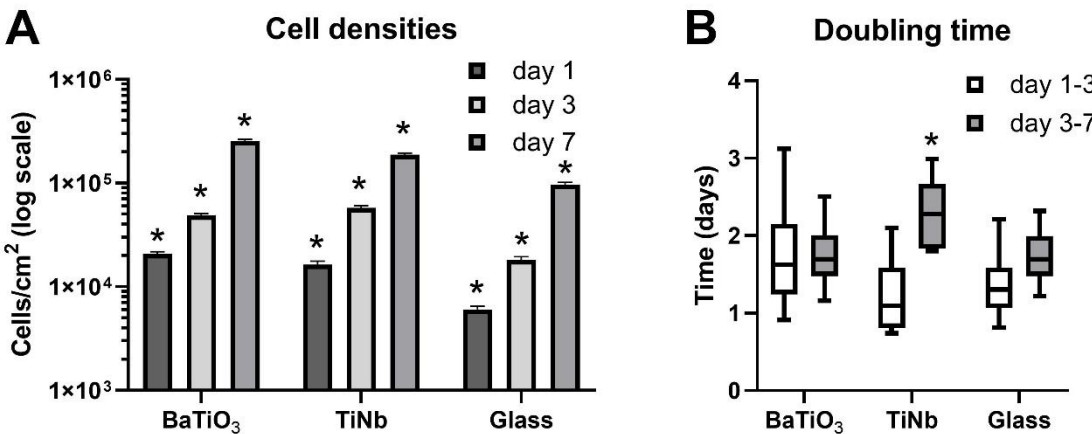

**Figure 8.** (**A**) Number of human osteoblast-like Saos-2 cells on day 1, 3 and 7 after seeding on Ti39Nb alloy coated with BaTiO$_3$ (BaTiO$_3$), on bare Ti39Nb alloy (TiNb), and on control microscopic glass coverslips (Glass). Mean + S.E.M. from three independent samples (in total from 48 to 52 microphotographs for each experimental group and time interval). ANOVA, Student-Newman-Keuls Methods. Asterisks above the columns indicate experimental groups significantly differing ($p \leq 0.05$) in cell number from all other groups on the same day of the culture. (**B**) Cell population doubling times of Saos-2 cells on the same samples, calculated between days 1 and 3 or between days 3 and 7 after seeding. Tukey boxplots, $n = 24$ for each plot. Bars within the boxes indicate medians; boxes indicate interquartile ranges (IQR); whiskers indicate the ranges of measurements within $1.5 \times$ IQR. Asterisks above the plots indicate experimental groups significantly differing from all other groups on the same day of the culture (ANOVA, * $p \leq 0.01$).

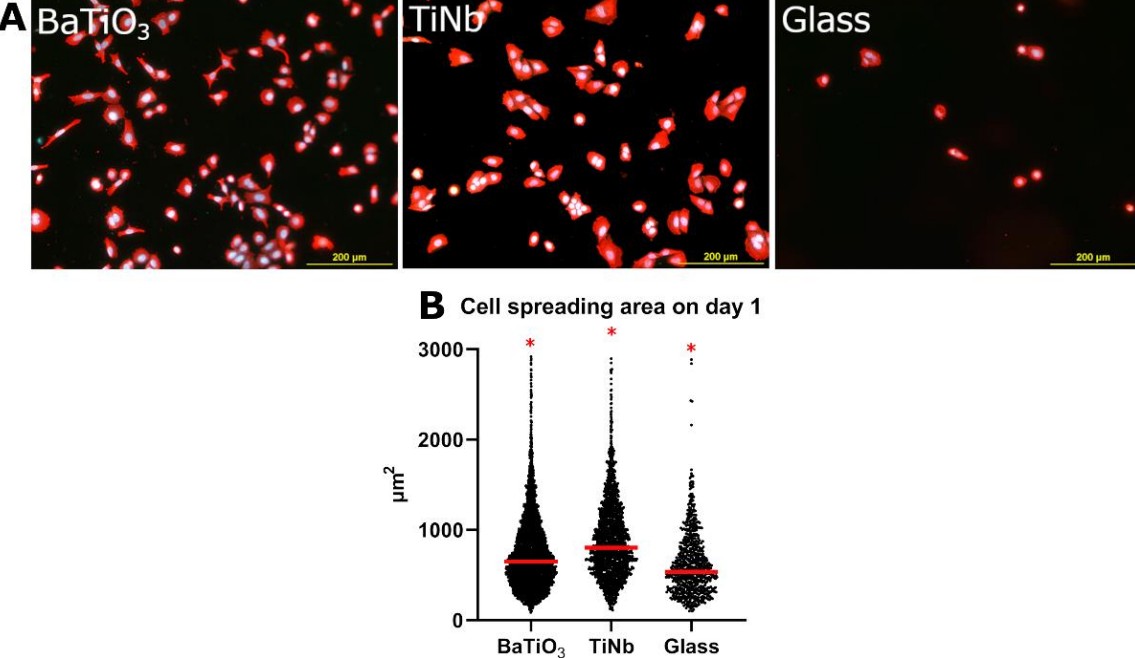

**Figure 9.** The morphology (**A**) and the violin plots showing the distributions of the cell spreading areas (**B**) of human osteoblast-like Saos-2 cells after 24 h in culture on Ti39Nb alloy coated with BaTiO$_3$ (BaTiO$_3$), on bare Ti39Nb alloy (TiNb), and on control microscopic glass coverslips (Glass). (**A**): The cells were fixed with the 70% frozen ethanol and stained with a combination of Hoechst #33258 (blue fluorescence) and Texas Red C$_2$-maleimide (red fluorescence). Scale bar = 200 μm. (**B**): The violin plot from 708 to 2510 cells for each experimental group, obtained from two samples for each group. ANOVA, Kruskal-Wallis, and Dunn's Methods. Red lines show median. Asterisks above the plots indicate experimental groups significantly differing from all other groups (* $p \leq 0.05$) in the size of cell spreading area.

Since the appropriate cell spreading is necessary for effective subsequent cell proliferation, the less-spread cells on $BaTiO_3$-coated samples proliferated more slowly, as manifested by their longer cell population doubling time (Figure 8B), and on day 3 after seeding, they attained a lower cell population density than the cells on the bare alloy (Figure 8A). However, there is also a possibility that the slower cell growth is due not only to a higher material surface roughness, but also to a release of $Ba^{2+}$ ions from the $BaTiO_3$ coating and by cytotoxic effect of these ions. This explanation was suggested in an earlier study performed on human osteoblast-like MG-63 cells cultured on $BaTiO_3$ coating deposited on a Ti6Al4V alloy [15]. The activity of mitochondrial enzymes in MG-63 cells, measured by MTT test and considered as an indirect indicator of cell proliferation, did not change during 7-day cultivation on $BaTiO_3$ coating, while in cells cultured on control bare Ti6Al4V, the activity increased from 1 to 7 days of cultivation [15]. The release of Ba from $BaTiO_3$ coating was also confirmed in the present study, and its negative effect on the cell proliferation cannot be excluded. Barium is known as a classic permeant blocker of potassium channels [54], which are implicated in proliferation of many cell types including osteoblasts [55,56]. In a study by Ciofani et al. (2010), the number of H9C2 rat cardiomyocytes in cultures exposed to $BaTiO_3$-containing polylysine nanoparticles decreased with increasing number of these nanoparticles [57]. In addition, fibroblasts from human periodontal ligament and human SCC9 keratinocytes cultured on poly(vinylidene fluoride–trifluoroethylene)/barium titanate membranes showed an increased expression of apoptosis-related genes (Bax, Bcl-2, and survivin) than the cells grown on reference polytetrafluoroethylene membranes [58]. In contrast, when 11 heavy metals including Ba, acting as food contaminants, were investigated for their potential genotoxicity (using $\gamma$H2AX assay detecting the DNA damage in human HepG2 and LS-174 cells), no significant genotoxicity of Ba was detected in comparison with the other metals [58]. In our study, barium also seemed not to act cytotoxically, although the leaching assay in the physiological saline solution confirmed the release of Ba from our $BaTiO_3$ coating (Figures 6 and 7). As indicated by the LIVE/DEAD assay, the viability of cells adhering to $BaTiO_3$-coated Ti39Nb samples was very high (almost 99%) and similar as on control bare samples and on reference glass coverslips (Table S1 in the Supplementary Materials). This finding is also in accordance with a study by Ball et al. (2014), where the barium titanate foams did not elicit any significant inflammatory response, evaluated by the release of TNF-$\alpha$ from human monocytic THP-1 cells [59].

Moreover, on day 7 after seeding, the cells on $BaTiO_3$ coating reached again a higher population density than on the bare Ti39Nb alloy (Figure 8A). This could be explained by a partial degradation of $BaTiO_3$ and covering the irregularities by extracellular matrix deposited by cells. On flatter coating, the cells were allowed to proliferate faster, as indicated by a shorter cell population doubling time between days 3 and 7 after seeding (Figure 8B). Similar results were obtained in a study performed on MG-63 cells cultured on highly porous barium titanate scaffolds. When these scaffolds were coated with a composite of gelatin and nanostructured hydroxyapatite, the cells proliferated more rapidly and reached higher population densities than on the bare scaffolds [60]. Covering $BaTiO_3$ with an additional layer can also reduce the release of Ba from this material. As a bioactive material, $BaTiO_3$ promoted deposition of calcium phosphates on its surface in a simulated body fluid, and this coverage then slowed down the Ba release [15]. Similarly, the coverage with cells also reduced the release of Ba from $BaTiO_3$ film, as demonstrated in our study (Figure 7).

However, the improved growth of Saos-2 cells on $BaTiO_3$ coating can be also attributed to a direct positive effect of this ferroelectric material on cells. $BaTiO_3$ coating contains positively and negatively charged sites on their surface, which, in addition to their beneficial effect on the adsorption of the cell adhesion-mediating proteins, may facilitate many other cellular processes. These processes include the mitochondrial activity, proteosynthesis, movement of charged molecules inside and outside the cell, and activation of ion channels in the cell membrane, such as Na, K, Cl and particularly Ca channels, which play important role in many vital functions of cells [18,52].

The adhesion and growth of cells on ferroelectric materials can be further improved by material poling, i.e., by inducing positively-charged sites on one surface and negatively-charged sites on the opposite surface. For example, when poled surfaces were introduced on ferroelectric potassium sodium niobate ceramics, the number of mouse MC3T3-E1 pre-osteoblasts adhering to these surfaces was higher than on unpoled surfaces [21]. A higher attachment, spreading, and subsequent proliferation was also described in MC3T3-E1 cells on composites of PLGA mixed with poled gadolinium-doped $BaTiO_3$ nanoparticles [28], and in rat bone marrow mesenchymal stem cells on poled $BaTiO_3$ coatings deposited on porous Ti6Al4V scaffolds [18].

### 3.7. Specific Markers of Adhesion and Differentiation in Cells on the Materials

The cell adhesion on both $BaTiO_3$-coated and uncoated Ti39Nb samples was investigated more deeply by immunofluorescence of $\beta_1$-integrins and vinculin, important molecules participating in cell adhesion. The integrins with $\beta_1$ chain include cellular receptors for main extracellular matrix proteins, such as collagen, an important component of the bone matrix ($\alpha_1\beta_1$, $\alpha_2\beta_1$, $\alpha_3\beta_1$ integrins), laminin, present in the basal laminae of cells ($\alpha_6\beta_1$, $\alpha_7\beta_1$ integrins) and also vitronectin and fibronectin, i.e., proteins of the serum supplement of the culture medium mediating the cell adhesion to biomaterials in vitro ($\alpha_5\beta_1$ and $\alpha_v\beta_1$ integrins). Vinculin, together with paxillin, talin, or $\alpha$-actinin, is a structural protein associated with integrin adhesion receptors after their recruitment into focal adhesion plaques. It occurs in relatively mature focal adhesion plaques, it stabilizes them and is involved in transduction of signals from the integrins to the actin cytoskeleton and to other intracellular structures, including cell nucleus [52].

As estimated by the intensity of immunofluorescence, on day 3 after seeding, the amount of $\beta_1$-integrins and vinculin was significantly higher in cells on both $BaTiO_3$-coated and uncoated Ti39Nb samples than in cells on reference microscopic glass coverslips. At the same time, the amount of $\beta_1$-integrins and vinculin was higher in cells on bare Ti39Nb samples than in $BaTiO_3$-coated samples (Figure 10). In accordance with this finding, the vinculin-containing focal adhesion plaques were more apparent in cells on the bare alloy (Figure 11). These results are in line with the largest spreading areas of cells on the bare Ti39Nb alloy on day 1 after seeding (Figure 9). It is known that vinculin is indispensable for the spreading, which is one of the manifestations of the cell motility, together with the cell migration [52].

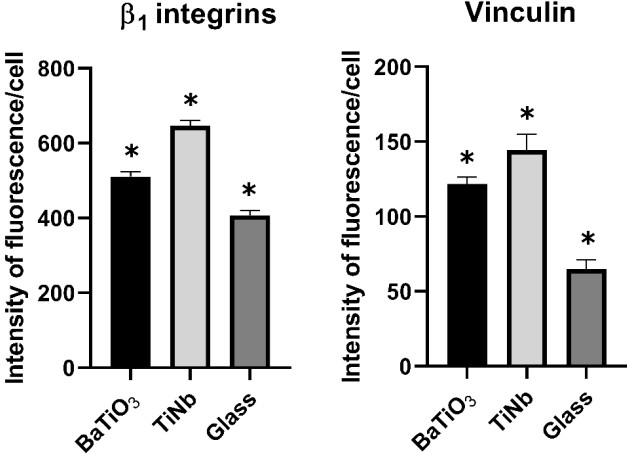

**Figure 10.** Intensity of immunofluorescence of molecules involved in cell adhesion, i.e., $\beta_1$-integrin adhesion receptors and vinculin, normalized to the cell number on the Ti39Nb alloy coated with $BaTiO_3$ ($BaTiO_3$), on bare Ti39Nb alloy (TiNb), and on control microscopic glass coverslips (Glass). Mean + S.E.M. from 16 microphotographs for each experimental group. ANOVA, Student-Newman-Keuls Methods. Asterisks above the columns indicate experimental groups significantly differing (* $p \leq 0.05$) from the other groups.

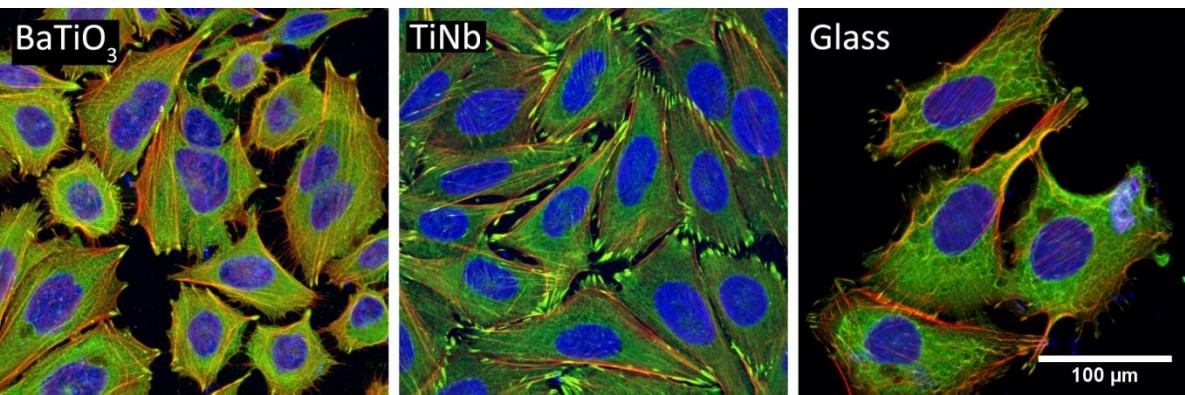

**Figure 11.** Immunofluorescence staining of vinculin (**green**), F-actin cytoskeleton (**red**), and nuclei (**blue**) in osteoblast-like Saos-2 cells on day 3 after seeding on Ti39Nb alloy coated with BaTiO$_3$ (BaTiO$_3$), bare Ti39Nb alloy (TiNb), and control microscopic glass coverslip (Glass). Bar = 100 μm.

On day 7, markers of osteogenic differentiation, namely ALP, i.e., an enzyme involved in the bone matrix mineralization, and collagen I, a major component of the bone matrix, were also evaluated by measuring the intensity of immunofluorescence. The results were similar as for the markers of cell adhesion—the amount of both differentiation markers was higher in cells on both coated and uncoated Ti39Nb samples than in cells on reference glass coverslips. At the same time, the amount of ALP was similar in cells on both coated and uncoated samples, but the amount of collagen I was significantly higher in cells on bare Ti39Nb than on BaTiO$_3$-coated samples (Figure 12). This result was rather surprising, because earlier studies reported a positive effect of ferroelectric BaTiO$_3$ on osteogenic cell differentiation. For example, in our earlier study, the intensity of fluorescence of ALP and osteocalcin, another marker of osteogenic cell differentiation, was significantly higher in Saos-2 cells on ferroelectric than on non-ferroelectric BaTiO$_3$ prepared on silica substrates with Pt interlayers [17]. In a recent study by Amaral et al. (2020), BaTiO$_3$ nanoparticles, added into hydrogel matrices prepared for bone tissue engineering, induced the expression of bone ALP in human dental pulp cells, while the same effect was not seen in the pure matrices [31]. Incorporation of BaTiO$_3$ nanoparticles into electrospun nanofibrous PLLA scaffolds also promoted osteogenic differentiation of bone marrow mesenchymal stem cells in cultures on these scaffolds [29].

Similar to the effect on cell proliferation, the effect of ferroelectric materials on osteogenic cell differentiation can also be modulated by poling these materials. In our earlier study performed on poled ferroelectric single crystalline LiNbO$_3$ plates, human osteoblast-like Saos-2 cells showed a higher activity of ALP on positively-poled surfaces than on unpoled surfaces [20]. However, the negatively-poled surfaces also hold a great promise in bone tissue engineering. As proved in an electroactive biocomposite of PLGA mixed with gadolinium-doped BaTiO$_3$ nanoparticles, these surfaces attracted positively-charged Ca$^{2+}$ ions, which stimulated the osteogenic differentiation of mouse MC3T3-E1pre-osteoblasts [28].

The positive effect of poling on the growth and osteogenic differentiation of cells can be further enhanced by mechanical loading, which results in manifestations of the piezoelectricity of ferroelectric materials [22]. Piezoelectric effect in BaTiO$_3$ was induced by gravity [26], ultrasound [61], by mechanical loading in a dynamic cell culture system simulating a human motion [24] or by mechanical stimulation after its implantation in vivo into experimental animals [18]. This piezoelectric BaTiO$_3$ showed osteoinductive and osteoconductive effects and an improved osseointegration.

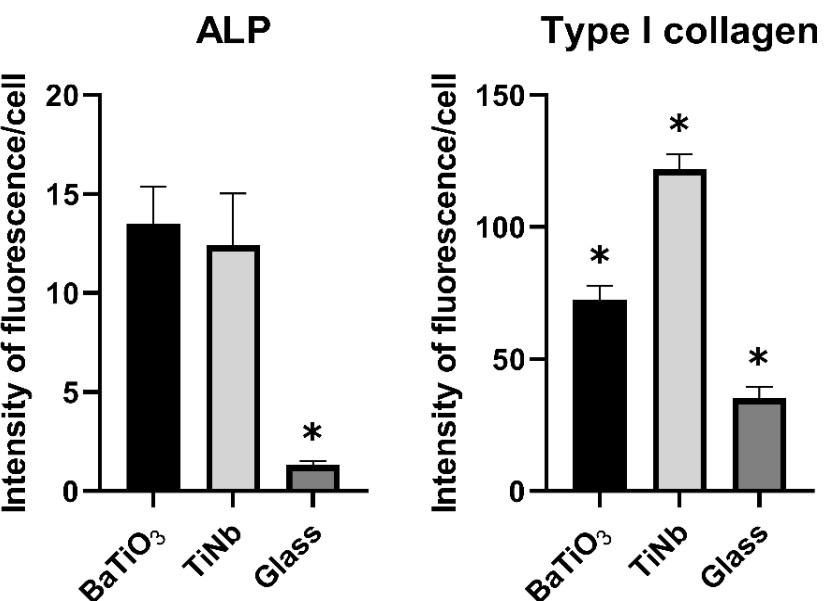

**Figure 12.** Evaluation of fluorescence intensity of molecules involved in osteogenic cell differentiation, i.e., alkaline phosphatase (ALP) and type I collagen, normalized to the cell number on the Ti39Nb alloy coated with BaTiO$_3$ (BaTiO$_3$), on bare Ti39Nb alloy (TiNb), and on control microscopic glass coverslips (Glass). Mean + S.E.M. from 16 microphotographs for each experimental group. ANOVA, Student-Newman-Keuls Methods. Asterisks above the columns indicate experimental groups significantly differing (* $p \leq 0.05$) from the other groups.

## 4. Conclusions

A ferroelectric coating using BaTiO$_3$ film was successfully prepared by hydrothermal synthesis on Ti39Nb alloy, i.e., a β-titanium alloy intended for construction of bone implants. The BaTiO$_3$ film was polycrystalline with random orientation of grains and crystallites, displaying ferroelectricity and piezoelectricity The BaTiO$_3$ film released Ba during leaching in the physiological saline solution and also during cell cultivation in vitro. Nevertheless, the cultivation of human osteoblast-like Saos-2 cells on BaTiO$_3$ film showed no cytotoxicity of this coating, as suggested by the cell viability higher than 98%, and also by other well-preserved cell functions, such as assembly of vinculin-containing focal adhesion plaques, formation of actin cytoskeleton, and subsequent cell growth. A lower cell spreading, slower initial proliferation, and a lower content of β$_1$-integrins, vinculin, and collagen I in cells on BaTiO$_3$ coating can be attributed to a higher surface roughness of this coating, which was in submicron-/micron-scale, while the roughness of the bare alloy was in nanoscale. In later culture intervals, when the BaTiO$_3$ film was partially degraded, the cells on this film accelerated their proliferation and reached even a higher cell population density than on the bare alloy. In addition, these cells were able to produce similar amount of alkaline phosphatase, an enzyme involved in osteogenesis, as the cells on the bare alloy. Thus, after improvements of the surface morphology and stability of BaTiO$_3$ films, and also after excitation of piezoelectric effect in these films by mechanical stimulation, these films will be promising candidates for coating metallic bone implants in order to improve their osseointegration.

**Supplementary Materials:** The following are available online at https://www.mdpi.com/2079-6412/11/2/210/s1, Figure S1: Representative images of human osteoblast-like Saos-2 cells, Table S1: Percentage of viable cell in cultures of human osteoblast-like Saos-2 cells in 1 day-old cultures on the tested surfaces.

**Author Contributions:** Conceptualization, M.V., P.V. and L.B.; methodology, M.V., Z.T., P.V., V.N., M.D., M.T., J.D., E.B., F.B. and J.N.; software, M.D., J.D., V.N. and F.B.; validation, M.V., Z.T., P.V., M.D., M.T., J.D. and V.N.; formal analysis, M.V., Z.T., P.V., M.D., M.T., J.D. and V.N.; investigation, M.V., Z.T., P.V., V.N., J.D., E.B., F.B. and J.N.; resources, M.V., Z.T., P.V., V.N. and J.D; data curation, M.V., Z.T., P.V., V.N. and J.D.; writing—original draft preparation, M.V., Z.T., P.V. and V.N.; writing—review and editing, M.V., P.V., F.B. and L.B.; visualization, M.V., Z.T., V.N., M.D., M.T., J.D., F.B., J.N. and L.B.; supervision, P.V., L.B.; project administration, M.V., P.V. and L.B.; funding acquisition, M.V., P.V. All authors have read and agreed to the published version of the manuscript.

**Funding:** This research was funded by the Czech Science Foundation (grant No. 20-01570S).

**Data Availability Statement:** The data presented in this study are available in supplementary.

**Conflicts of Interest:** The authors declare no conflict of interest. The funders had no role in the design of the study; in the collection, analyses, or interpretation of data; in the writing of the manuscript, or in the decision to publish the results.

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
