# Peer review of "Beta-Titanium Alloy Covered by Ferroelectric Coating–Physicochemical Properties and Human Osteoblast-Like Cell Response"

_coatings, doi:10.3390/coatings11020210_

Round 1
Reviewer 1 Report
The manuscript no. 1060726 entitled: „Human osteoblast-like cells on ferroelectric coatings prepared on a beta-titanium alloy” presents synthesis and evaluation of physicochemical as well as biological properties of new biomaterials dedicated for bone tissue engineering.
The article is very interesting and will be appropriate to publish in Coatings journal, when some minor remarks will be considered:
General remark
The authors use the term “ferroelectric coatings”; “films” etc. How many types of coatings they prepared? I see only one type, namely BaTiO3 coating on Ti39Nb. If it is only one type of coating – the reviewer believe that will be better to use the singular form (coating, film etc.) in order to not mislead the readers. If this coating is composed of few layers – for example – the authors may use the term “multi-layer BaTiO3 coating” or other. This issue should be verified and corrected in whole body of manuscript.
Detailed remarks
- Title
I suggest to change the Title of presented manuscript. The Article includes not only cell culture experiments in vitro, but also physicochemical characterization of beta-titanium alloy.
Suggested title:
“Beta-titanium alloy covered by ferroelectric coating – Physicochemical characterization and evaluation of human osteoblast-like cell response”.
- Introduction
Lines 39-54 and 76-81
The mentioned paragraphs should be fulfill by adequate references. Please, add suitable positions of literature.
- Results and Discussion
3.1. Basic properties of the BaTiO3 films
Lines 330-332
The authors wrote: ”It is apparent that the roughness of the BaTiO3 layers is markedly higher than that of the original Ti39Nb substrates. The roughness of these layers is in submicron-/micron-scale, while the roughness of the bare Ti39Nb substrates is in nanoscale”.
Please, add statistical data and then use the term “markedly higher”.
Lines 340-341
The authors wrote: “The shape and dimensions of the crystals (together with their chemical composition) have significant influence on the ferroelectricity of the material surface”.
Please, expand. The term “significant influence” is too general. If it is well-described knowledge, please introduce adequate references.
3.2. XRD analysis of the Ti39nb substrates and BaTiO3films
Lines 347-348
The authors wrote: “The Im-3m cubic phase has the same space group and 347 a larger lattice parameter (3.35 Å) than the original Ti39Nb phase (3.28 Å)”.
The same space group (…) than the original or the same space group (…) as the original. Please, correct.
Figure 3.
Please, correct. It is visible only partially.
Author Response
Responses to Reviewer 1
We would like to thank the reviewer for useful comments, which helped us considerably to improve our manuscript. The revisions are highlighted in blue in the manuscript, and our responses in this letter are written in red.
General remark
The authors use the term “ferroelectric coatings”; “films” etc. How many types of coatings they prepared? I see only one type, namely BaTiO3 coating on Ti39Nb. If it is only one type of coating – the reviewer believe that will be better to use the singular form (coating, film etc.) in order to not mislead the readers. If this coating is composed of few layers – for example – the authors may use the term “multi-layer BaTiO3 coating” or other. This issue should be verified and corrected in whole body of manuscript.
Response: The reviewer is right that we have prepared and investigated only one type of coating, namely BaTiO3. Thus, now we use the singular form (i.e. film, coating) throughout the whole manuscript, where it is relevant. (Due to a high number of these changes, they are not always highlighted.)
- Title
I suggest to change the Title of presented manuscript. The Article includes not only cell culture experiments in vitro, but also physicochemical characterization of beta-titanium alloy.
Suggested title:
“Beta-titanium alloy covered by ferroelectric coating – Physicochemical characterization and evaluation of human osteoblast-like cell response”.
Response: The Reviewer 1 is right that the physicochemical characterization of the material and in vitro cell response to it are balanced, and thus the new title proposed by the Reviewer is a good idea. We only slightly modified the suggestion of the Reviewer 1. The new title is:
“Beta-titanium alloy covered by ferroelectric coating – physicochemical properties and human osteoblast-like cell response”.
- Introduction
Lines 39-54 and 76-81
The mentioned paragraphs should be fulfill by adequate references. Please, add suitable positions of literature.
Response: Adequate references have been added to the mentioned paragraphs. For both paragraphs, we have added new references (Gabor et al. 2020, Vandrovcova et al. 2011, Stepanovska et al. 2020). In addition, for the second paragraph, we have also utilized suitable references already cited in the manuscript, i.e. Rahmati et al. 2016, Jelinek et al. 2018, Tolde et al. 2017, Liu et al. 2020), dealing with electroactive materials.
- Results and Discussion
3.1. Basic properties of the BaTiO3 films
Lines 330-332
The authors wrote: ”It is apparent that the roughness of the BaTiO3 layers is markedly higher than that of the original Ti39Nb substrates. The roughness of these layers is in submicron-/micron-scale, while the roughness of the bare Ti39Nb substrates is in nanoscale”.
Please, add statistical data and then use the term “markedly higher”.
Response: We added the statistical data for the roughness of the substrate and of the BaTiO3 film. The statistical significance of the difference in the Ra parameter of both materials was evaluated by ANOVA (Student-Newman-Keuls Method), and also by the t-test. Both approaches have indicated that the Ra parameter is significantly higher (p≤0.001) in the BaTiO3 film. It is worth mentioning that the average Ra parameter is more than 60 times higher in the BaTiO3 film (0.931 mm) than in the bare TiNb alloy (0.015 µm).
Lines 340-341
The authors wrote: “The shape and dimensions of the crystals (together with their chemical composition) have significant influence on the ferroelectricity of the material surface”.
Please, expand. The term “significant influence” is too general. If it is well-described knowledge, please introduce adequate references.
Response: This paragraph has been modified, extended and an adequate reference has been added. In addition, for a better discussion of the obtained results in context with XRD, the paragraph has been moved to the chapter 3.2. The new formulation of the paragraph is as follows:
“It is known that the dimensions of the crystallites (size effect) together with their chemical composition have a significant influence on the development of ferroelectricity on the material surface [Zhao et al. 2004]. In our case, the size of the crystallites determined by Rietveld refinement of XRD was about 40 nm, which allows the development of stable spontaneous polarization, although ferroelectricity can be weaker than in bigger crystals. It is also obvious that the most of the crystal grains seen by SEM (Figure 1D) are composed of several crystallites”.
3.2. XRD analysis of the Ti39nb substrates and BaTiO3films
Lines 347-348
The authors wrote: “The Im-3m cubic phase has the same space group and 347 a larger lattice parameter (3.35 Å) than the original Ti39Nb phase (3.28 Å)”.
The same space group (…) than the original or the same space group (…) as the original. Please, correct.
Response: The mentioned sentence has been modified as follows:
“The Im-3m cubic phase has the same space group but a larger lattice parameter (3.35 Å) compared with the original Ti39Nb phase (3.28 Å)”.
Figure 3.
Please, correct. It is visible only partially.
Response: It was probably an error of the Word version of the manuscript. It has been corrected.
Reviewer 2 Report
After completing the paper, Human osteoblast-like cells on ferroelectric coatings prepared on a beta-titanium alloythe, first impression is that the authors managed to build a well-documented article and supported by a consistent experimental part. In addition, the exhaustive approach of the biology part seems to crown the existence of an exceptional result.
However, even taking into account the final conclusions, elements that raise key questions reveal the weaknesses of the article as follows:
- the evidence regarding the formation of a barium titanate layer is not convincing, although a pretentious characterization is used;
- the surface morphology is not convincing, because it does not prove the existence of a correctly developed layer in 2D;
- there is no evidence to provide the stoichiometry; it is known that this aspect essentially modifies the quality of barium titanate;
- no evidence of piezoelectricity, although almost the entire article is based on the existence of this property;
- the authors must also consult other published works in which they will find useful information about the formation and properties of barium titanate by the hydrothermal method;
- the authors compare properties of bulk barium titanate with those of 2D; must understand how the properties of materials change, especially those with intrinsic polarization, depending on the size taking into account the aspects of DOS theory.
In order to be published, the authors must come up with evidence regarding the existence of the barium titanate layer and its electrical properties.
Otherwise, the whole plea of the paper remains without the fundamental support, ie the existence of a layer of barium titanate grown uniformly, stoichiometry and stable in which the properties desired by the authors are manifested.
The recommendations are as follows:
- highlighting the stoichiometry of the target compound;
- section view of the deposited layer with appropriate methods (UHR SEM with EDS, TEM with HRTEM etc ;,
- highlighting the piezolelectric coefficient;
- more experimental results.
In conclusion, the proposal is for the revision of the paper with the Major Revision level.
Author Response
Responses to Reviewer 2
We would like to thank the reviewer for useful comments, which helped us considerably to improve our manuscript. The revisions are highlighted yellow in the manuscript, and our responses in this letter are written in red. If the same revision is also recommended by the Reviewer 1, blue color is used for highlighting.
Comments:
However, even taking into account the final conclusions, elements that raise key questions reveal the weaknesses of the article as follows:
- the evidence regarding the formation of a barium titanate layer is not convincing, although a pretentious characterization is used;
- the surface morphology is not convincing, because it does not prove the existence of a correctly developed layer in 2D;
- there is no evidence to provide the stoichiometry; it is known that this aspect essentially modifies the quality of barium titanate;
- no evidence of piezoelectricity, although almost the entire article is based on the existence of this property;
- the authors must also consult other published works in which they will find useful information about the formation and properties of barium titanate by the hydrothermal method;
- the authors compare properties of bulk barium titanate with those of 2D; must understand how the properties of materials change, especially those with intrinsic polarization, depending on the size taking into account the aspects of DOS theory.
In order to be published, the authors must come up with evidence regarding the existence of the barium titanate layer and its electrical properties.
Otherwise, the whole plea of the paper remains without the fundamental support, ie the existence of a layer of barium titanate grown uniformly, stoichiometry and stable in which the properties desired by the authors are manifested.
The recommendations are as follows:
- highlighting the stoichiometry of the target compound;
- section view of the deposited layer with appropriate methods (UHR SEM with EDS, TEM with HRTEM etc ;,
- highlighting the piezoelectric coefficient;
- more experimental results.
Responses:
Ad weaknesses 1, 3, 4 and recommendations 1, 3, 4:
In order to show the composition of the film and its ferro/piezoelectric character, we have made the following amendments and improvements, including a higher number of characterization methods:
- We have added EDS analysis with new Fig. 2 showing the chemical composition throughout the film.
- We have added Fig. 3C comparing experimental XRD data with the theoretical fit of the BaTiO3 structure. These data prove that the deposited film is composed of BaTiO3.
- The chapter 3.3. Raman spectroscopy of Ti39Nb substrates and BaTiO3 film has been extended, and two new references have been added. The chapter now confirms the presence of ferroelectric (and also piezoelectric) BaTiO3 in the deposited film.
- A new chapter 3.4. Piezoresponse force microscopy (PFM) of BaTiO3 film has been added including experimental results which confirm the piezoelectricity and ferroelectricity of the deposited film.
- The chapter 3.5. XPS analysis of BaTiO3 film on Ti39Nb substrates shows the stoichiometric ratios of BaTiO3 in the surface layer of the film.
Ad weaknesses 2, 6 and recommendations 2, 4:
The SEM and HRTEM pictures of the cross section of the substrate coated by the film have been added (Fig. 1 E, F). The BaTiO3 film has a thickness of about 1 µm, therefore it cannot be regarded as 2D material but rather as a bulk.
Ad weakness 5:
We took the information on hydrothermal method from several papers. The references [18] (Liu et al. 2020) and [39] (Zhu et al. 1998) mentioned in the present version of the manuscript are the most relevant for our research.
Reviewer 3 Report
In this study, the substrate made of the beta-titanium alloy has been coated with barium titanate to make it more favourable to be used as a bone implant. Authors characterised the coating with various techniques, such as; scanning electron microscopy and Raman microscopy, X-ray diffraction, X-ray photoelectron spectroscopy, nanoindentation, and roughness measurement. The biological performance of the coating has been assessed by culturing Saos-2 cells on the material-as received, the coated substrate, and the glass as a control.
A great variety of techniques have been used for physical, chemical, and biological assessment of the material. However, there are major and minor points that need to be either clarified or revised before re-consideration of the manuscript.
- One of the main drawbacks in the manuscript is that the n numbers (number of samples) are confusing and insufficient. Please report n numbers for each assessment in the figure legends of all the graphs.
- What is N (number of repeats) number? Have all the experiments been conducted only once (without repeat)?
- As far as I am concerned, the authors used only one sample from each group (which means n=1), for LIVE/DEAD assay, and there is not any repeat (N=1). Although they also used 16 microscopic fields from each sample for the quantification and statistical tests, it is not acceptable. There must be at least three samples from each group for the reliability of the biological assessments.
- Was live&dead assay conducted only on day 1? If so, how have the cell densities been calculated and reported in Figure 7? Please better clarify which biological analysis was conducted on which day and indicate the name of the analysis in the caption of the corresponding figure. For example; for Figure 7, which analysis has been implemented?
- In Figure 11, the F-actin of the cells was stained with red. Throughout the text, it has not been mentioned which dye has been used for staining the cytoskeleton.
- In Table 1, only mechanical properties of the coating layer have been reported. Please also report the mechanical properties of the substrate for comparison.
- Figure 6, please add error bars and n numbers.
- Please support the following statement (Line 487) with references;
“However, a higher material surface roughness is advantageous only for the initial cell attachment, when the seeded cells, suspended in the culture medium, contact the material in their round shape. When the cells start to spread on the material surface, i.e. to increase their area contacting the material, the irregularities on the material surface may hamper this process. The cells usually try to spread over tens of micrometers, and irregularities in micrometer scale, present on the BaTiO3 films may be limiting for this process.”
- Please provide the representative micrographs for live&dead assay as supplementary material.
- The Introduction and Results&discussion Sections are too long. Please shorten them significantly.
- Please refrain from over-explaining the details of studies from the literature (Introduction&Discussion). Only report the related conclusion sentence about the study and provide the reference for the reader who is interested in further details.
- Please remove the phone number from correspondence and report only the e-mail address of the corresponding author instead.
- Do not you think the following two statements from the manuscript are conflicting?
(Line 525) “However, the slower cell growth could be explained not only by a higher material surface roughness, but also by leaching of Ba2+ ions from the BaTiO3 coatings and by cytotoxic effect of these ions.”
(Line 545) “In our study, barium also seemed not to act cytotoxically, although the leaching assay in the physiological saline solution confirmed the release of Ba from our BaTiO3 coatings (Figures 5, 6). As indicated by the LIVE/DEAD assay, the viability of cells adhering to BaTiO3-coated Ti39Nb samples was very high (almost 99%) and similar as on control bare samples and on reference glass coverslips.”
- Please, always use the fulls names (abbreviations in the parenthesis) of elements and compounds in their first use in the manuscript and keep using the abbreviated form in the rest of the manuscript.
- Most of the sentences in the Introduction section is lack of references. Please support these sentences with related references from the literature.
-especially the ones in the earlier paragraphs-
- Throughout the text, please remove “(for a review, see [REF])”. Instead, only give the reference.
- Please report the product details of HCl.
- Please replace “ml” with “mL” throughout the text.
- Please report if 20% acetic acid is volume-based and add the brand (Line 179).
- 70% ethanol treatment is not a sterilisation method. It is a disinfection method. Please replace it with “disinfection” (Section 2.6).
- After ethanol treatment, were the substrates air dried or washed with PBS or water or directly transferred to culture? Please provide further details.
- Please report the frequency of the media change.
- Please report the surface areas or the diameters of the glass coverslips and the substrates (BaTiO3+ Ti39Nb and Ti39Nb) used for cell culture experiments to give an insight about cell seeding density (Section 2.6).
- On day 1, were samples transferred to a fresh well plate, before LIVE/ DEAD assay?
- Please provide a reference for the method used in Section 2.8.
- Table 1, please keep the number of decimals consistent for each parameter (0.930±047 or 0.93±0.05).
- Figure 1, please crop the bottom of the images, report the process parameters in the Methods section and add the scale bars manually on the images for their better visibility.
- Please combine the graphs in Figure 7A, Figure 7B, and Figure 7C in one graph for better comparison and also show the cell densities of each group for different time points together. First, plot the cells densities on BaTiO3 at day 1, 3, 7, then for TiNb and finally for glass in the same graph.
- Please report doubling time in one graph (Figure 9A and Figure 9B).
- Figure 11, please add scale bars and remove the last sentence from the figure caption.
- Please do not report the details related to the materials&methods in figure captions. For example; - Figure 11, please remove “Leica SPE confocal microscope, Germany”.
- Please report Figure 7, Figure 9 and Table 3 in one figure (Figure XA, Figure XB).
- In conclusions, authors report that “However, the cultivation of human osteoblast-like Saos-2 cells on BaTiO3 layers showed a very low or none cytotoxicity of these coatings, because the cell viability was higher than 98%.”
Is it low or noncytotoxic? I would suggest applying a statistical test to data in Table 3 and reporting the cytotoxicity by comparing its significant difference with two other substrates.
Author Response
Revisions and responses to Reviewer 3
We would like to thank the reviewer for useful comments, which helped us considerably to improve our manuscript. The revisions are highlighted in green in the manuscript, our responses are written in red. If the same revision is also recommended by the Reviewers 1 and 2, blue and yellow colors are used for highlighting.
- One of the main drawbacks in the manuscript is that the n numbers (number of samples) are confusing and insufficient. Please report n numbers for each assessment in the figure legends of all the graphs.
Response:
For physicochemical analyses of the materials (Table 1 and Figures 1-7), from 1 to 5 samples for each experimental group were used, and on these samples, the analyses were made in 1 to 5 regions. In some analyses, particularly XRD, these regions were relatively large, comprising almost the whole sample surface (i.e. 0.79 cm2), or an area of 5x5 mm in XPS analysis. A similar approach is routinely applied in most studies dealing with biomaterials, and therefore we believe that the obtained results are reliable. The number of samples and number of analyzed regions has been specified in all the legends.
For biological analyses, 9 samples for each experimental group (i.e. bare TiNb, BaTiO3-coated TiNb and reference glass) were used. Three of them were used for analyses performed on day 1, three for day 3, and three for day 7. Therefore, the cell numbers were counted from three samples for each experimental group and time interval.
On day 1, the cells were counted the following 3 samples: one sample stained with LIVE/DEAD kit, which was also used for evaluation of cell viability, and the remaining two samples stained with a combination of Texas Red/Hoechst, which were also used for measuring the size of the cell spreading areas. On the LIVE/DEAD-stained sample, the cells were counted in 16 randomly distributed microscopic fields (microphotographs), and on the Texas Red/Hoechst-stained samples, the cells were counted on 18 randomly-selected fields per each sample (i.e., 36 fields in total). In total, the cell number was evaluated in 52 (16 + 36) randomly selected fields on each of the investigated surfaces.
On day 3, the cells were also counted on 3 samples. The cells on the one sample were stained for beta1-integrins, the cells on the second sample were stained for vinculin, and the cells on the third sample served as negative control without primary antibody. Nevertheless, the cell nuclei on all three samples were counterstained with Hoechst and the filamentous actin with TRITC-conjugated phalloidin, which enabled the cell counting on all three samples. On each sample, the cells were counted in 16 randomly selected fields, thus we evaluated 48 fields in total for each type of the investigated surfaces.
Similarly, on day 7, the cells were counted on three samples – one of them stained against alkaline phosphatase, the second against collagen I, the third used as a negative staining control. On all three samples, the cell nuclei were counterstained with Hoechst. Again, the cells on each sample were counted in 16 fields, i.e. in 48 fields in total for each experimental group.
Since the physical and chemical surface properties and the cell coverage were homogeneous on all tested surfaces, we believe that in spite of a relatively low number of samples, this number and the number of evaluated fields on each sample were sufficient for providing convincing, reliable results. The number of samples was limited by the size of the autoclave, in which the BaTiO3 samples were prepared, and particularly by a long-lasting preparation, the duration of which was 6 weeks.
The number of samples and number of evaluated fields on each sample were also added to the Material and Methods section (especially to the chapter 2.7. Evaluation of cell number and cell population doubling time), and the confusing parts of the previous original text were corrected and clarified.
- What is N (number of repeats) number? Have all the experiments been conducted only once (without repeat)?
Response: As mentioned above, for biological studies, three independent samples for each experimental group and time interval were used, and these samples were evaluated in 48-52 fields in total. These samples were utilized not only for cell counting, but also for analyses of cell viability, cell spreading area, arrangement and content of molecules participating in cell adhesion and spreading (beta1-integrins, vinculin, F-actin), and content of molecules participating in osteogenic cell differentiation, such as collagen type I and alkaline phosphatase. Although the number of analyzed samples was relatively low, and each analysis was conducted only once, this investigation enabled a complex insight into the cell behavior on the tested surfaces.
- As far as I am concerned, the authors used only one sample from each group (which means n=1), for LIVE/DEAD assay, and there is not any repeat (N=1). Although they also used 16 microscopic fields from each sample for the quantification and statistical tests, it is not acceptable. There must be at least three samples from each group for the reliability of the biological assessments.
Response: As mentioned above, we had three samples for each experimental group and time interval, but these samples were used for several analyses. The Referee is right that the LIVE/DEAD assay was performed only on one sample for each experimental group. For this reason, this analysis was moved to the Supplementary Material (Table S1, former Table 3) together with the newly added pictures of stained cells (Figure S1), also recommended by the Referee. However, it is clear from those pictures that the waste majority of cells is alive, and a similar picture was seen in all 16 microscopic fields throughout the entire sample. From this point of view, it is rather hard to imagine that even if we have more parallel samples, this picture would be considerably different. In addition, other results, i.e. increasing cell number from day 1 to day 7, good cell spreading, formation of vinculin-containing focal adhesion plaques and well-developed actin cytoskeleton and production of markers of osteogenic cell differentiation, also suggest a good cell viability on all investigated surfaces, including BaTiO3.
- Was live&dead assay conducted only on day 1? If so, how have the cell densities been calculated and reported in Figure 7? Please better clarify which biological analysis was conducted on which day and indicate the name of the analysis in the caption of the corresponding figure. For example; for Figure 7, which analysis has been implemented?
Response: The LIVE/DEAD assay was conducted on day 1 and the viability of the cells (i.e. the ratio of viable cells to all cells) was calculated on the basis of this assay. As mentioned above, for calculating the cell densities, also the samples stained with Texas Red/Hoechst (day 1), with immunofluorescence against beta-1 integrins and vinculin (day 3) and with immunofluorescence against collagen I and alkaline phosphatase (day 7) were used. The cell nuclei on the immunofluorescence-stained samples were counterstained with Hoechst, which enabled a precise cell counting, even on unstained control samples. By this way, for each experimental group and time interval, the cells were counted on three samples. This approach has been explained in the chapter 2.7. Evaluation of cell number and cell population doubling time). The numbers of samples and of microscopic fields on each sample have been added to the legend of each relevant figure.
- In Figure 11, the F-actin of the cells was stained with red. Throughout the text, it has not been mentioned which dye has been used for staining the cytoskeleton.
Response: We apologize for this omission – we have forgotten to mention it in the Material and Methods section of the original manuscript. This information (i.e. fluorescence staining with phalloidin conjugated with TRITC) has been added into the chapter 2.7. Evaluation of cell number and cell population doubling time).
- In Table 1, only mechanical properties of the coating layer have been reported. Please also report the mechanical properties of the substrate for comparison.
Response: The mechanical properties of the substrate have been added to the Table 1.
- Figure 6, please add error bars and n numbers.
Response: It is Figure 7 after revision. Error bars have been added; they mean 5 % of values presented in Figure, which is a usually supposed inaccuracy in XPS results.
As for the number of samples, the XPS analyses were made on one sample per experimental group in a field of 5x5 mm.
- Please support the following statement (Line 487) with references;
“However, a higher material surface roughness is advantageous only for the initial cell attachment, when the seeded cells, suspended in the culture medium, contact the material in their round shape. When the cells start to spread on the material surface, i.e. to increase their area contacting the material, the irregularities on the material surface may hamper this process. The cells usually try to spread over tens of micrometers, and irregularities in micrometer scale, present on the BaTiO3 films may be limiting for this process.”
Response: The statement has been supported with 3 references (review articles by Bacakova et al. 2011, already cited in the manuscript, a review article by Vandrovcova et al. 2011, and a recent paper by Steinerova et al. 2021. All these papers deal with the influence of the material surface roughness on the cell-material contact).
- Please provide the representative micrographs for live&dead assay as supplementary material.
Response: We have added the requested representative micrographs to Supplementary Materials, together with Table 3 (now Table S1)
- The Introduction and Results&discussion Sections are too long. Please shorten them significantly.
Response: These sections have been shortened, mainly by avoiding most of the texts dealing with the material poling, which is not directly related to our study. The considerably shortened parts of the Introduction and Results & Discussion sections are highlighted in green. Some figures were fused together. Some results (related to LIVE/DEAD assay) were moved to the Supplementary Materials. The number of original references has been reduced (in order to gain space for the newly added references required by the referees).
- Please refrain from over-explaining the details of studies from the literature (Introduction&Discussion). Only report the related conclusion sentence about the study and provide the reference for the reader who is interested in further details.
Response: The details of the studies from the literature have been avoided, if they are not directly related to our results, and only conclusion sentences with references have been provided.
- Please remove the phone number from correspondence and report only the e-mail address of the corresponding author instead.
Response: According to the journal guidelines we have to provide e-mail address of corresponding author, but according to the template, we may also include the phone number, which was optional. We have therefore decided to include both, and the decision which one would be used depends on the Editor.
- Do not you think the following two statements from the manuscript are conflicting?
(Line 525) “However, the slower cell growth could be explained not only by a higher material surface roughness, but also by leaching of Ba2+ ions from the BaTiO3 coatings and by cytotoxic effect of these ions.”
(Line 545) “In our study, barium also seemed not to act cytotoxically, although the leaching assay in the physiological saline solution confirmed the release of Ba from our BaTiO3 coatings (Figures 5, 6). As indicated by the LIVE/DEAD assay, the viability of cells adhering to BaTiO3-coated Ti39Nb samples was very high (almost 99%) and similar as on control bare samples and on reference glass coverslips.”
Response: We believe that these statements are not in conflict, because the first statement is only assumption, and the second statement is an argument against this expected cytotoxicity, arising from our results. Nevertheless, the first sentence has been more precisely formulated in order to avoid confusion.
- Please, always use the fulls names (abbreviations in the parenthesis) of elements and compounds in their first use in the manuscript and keep using the abbreviated form in the rest of the manuscript.
Response: The full names (abbreviations in the parenthesis) of elements and compounds in their first use have been added to the manuscript.
- Most of the sentences in the Introduction section is lack of references. Please support these sentences with related references from the literature.
-especially the ones in the earlier paragraphs-
Response: A similar concern has been raised by Reviewer 1, and therefore we have added new references especially to the first paragraph. Other paragraphs were enriched by suitable references already cited in the manuscript.
- Throughout the text, please remove “(for a review, see [REF])”. Instead, only give the reference.
Response: The phrase “for a review, see” was used for citing review articles in order to distinguish them from primary articles with original findings. However, this phrase has been removed throughout the entire text, and only the references have been given.
- Please report the product details of HCl.
Response: The product details of HCl were added.
- Please replace “ml” with “mL” throughout the text.
Response: We have replaced ml for mL throughout the text. Both l and L are now allowed by UPAP for liter. Earlier, only l was allowed by IUPAP for liter. Now L is allowed by IUPAP as the exception from the rule that units are marked by small letters. The reason is that l (liter) could be confused with 1 (one).
- Please report if 20% acetic acid is volume-based and add the brand (Line 179).
Response: We have changed 20 % acetic acid to 20 vol.% acetic acid, and the product details have been added.
- 70% ethanol treatment is not a sterilisation method. It is a disinfection method. Please replace it with “disinfection” (Section 2.6).
Response: We have replaced the word “sterilisation” with “disinfection” in the section 2.6.
- After ethanol treatment, were the substrates air dried or washed with PBS or water or directly transferred to culture? Please provide further details.
Response: After the ethanol treatment, the samples were washed with PBS, inserted into 24-well plates and seeded with Saos-2 cells. We have added this information into the section 2.6.
- Please report the frequency of the media change.
Response: The cells were cultivated on the materials for 7 days without any medium change. We believed that these changes could mask, at least partly, the influence of the investigated materials on cells. Even the osteogenic cell differentiation was not studied after change of the growth medium for a differentiation medium in order to the pure effect of the material on the phenotypic maturation of cells.
- Please report the surface areas or the diameters of the glass coverslips and the substrates (BaTiO3+ Ti39Nb and Ti39Nb) used for cell culture experiments to give an insight about cell seeding density (Section 2.6).
Response: The diameter of the glass coverslips was 12 mm (i.e. 1.1304 cm2) and the diameter of the substrates was 10 mm (i.e. 0.79 cm2). We have added this information to the chapter 2.6 Cell seeding.
- On day 1, were samples transferred to a fresh well plate, before LIVE/ DEAD assay?
Response: During the LIVE/DEAD assay the cells on samples inserted in the original well plate were fluorescently stained with two probes, which allows to distinguish between dead and living cells (please, view the methods for details). Then, the samples were removed from the well plate, were placed on glass coverslips and viewed under the inverted fluorescent microscope to determine the number of living and dead cells from microphotographs. The LIVE/DEAD assay means direct cell staining, and not colorimetric assay, i.e. assay where the reaction product is released into the cell culture medium, like in MTT. MTS. XTT or WST assays. Therefore, there is no need to transfer the samples to a fresh plate before performing the LIVE/DEAD assay, because this assay cannot be influenced by cells growing on the well bottoms under or around the samples.
- Please provide a reference for the method used in Section 2.8. “Evaluation of the intensity of fluorescence”
Response: The reference was provided in Section 2.8 (Filova et al. 2015)
- Table 1, please keep the number of decimals consistent for each parameter (0.930±047 or 0.93±0.05).
Response: Now the number of decimals is consistent for each parameter.
- Figure 1, please crop the bottom of the images, report the process parameters in the Methods section and add the scale bars manually on the images for their better visibility.
Response: Process parameters are now in the chapter 2.3. Scale bars were put manually on the images.
- Please combine the graphs in Figure 7A, Figure 7B, and Figure 7C in one graph for better comparison and also show the cell densities of each group for different time points together. First, plot the cells densities on BaTiO3 at day 1, 3, 7, then for TiNb and finally for glass in the same graph.
Response: We have combined the three graphs into one graph as requested.
- Please report doubling time in one graph (Figure 9A and Figure 9B).
Response: We have combined the two graphs into one graph as requested.
- Figure 11, please add scale bars and remove the last sentence from the figure caption.
Response: We have added the scale bar to the figure (Figure 13 in the revised manuscript) and modified its caption accordingly.
- Please do not report the details related to the materials&methods in figure captions. For example; - Figure 11, please remove “Leica SPE confocal microscope, Germany”.
Response: The respective figure captions (Figures 11 and 9) have been modified as requested. The specification of the microscopes was moved to the Material and Methods section.
- Please report Figure 7, Figure 9 and Table 3 in one figure (Figure XA, Figure XB).
Response: We have combined original Figure 7 and Figure 9 into one figure (i.e. Figure 9) in the revised manuscript. Table 3 has been moved to the Supplementary Materials.
- In conclusions, authors report that “However, the cultivation of human osteoblast-like Saos-2 cells on BaTiO3 layers showed a very low or none cytotoxicity of these coatings, because the cell viability was higher than 98%.”
Is it low or noncytotoxic? I would suggest applying a statistical test to data in Table 3 and reporting the cytotoxicity by comparing its significant difference with two other substrates.
Response: We have applied statistical tests (One Way ANOVA and ANOVA on Ranks), and these tests did not reveal any statistically significant difference in the cell viability on BaTiO3 film, bare TiNb and microscopic glass coverslip. The cell viability of 98.7% on the BaTiO3 film can be considered very high, and the small number of dead cells can be considered to be within the limit of physiological cell loss in normal cell populations. Thus, it can be concluded that BaTiO3 in our study acted as non-cytotoxic.
Round 2
Reviewer 1 Report
I accept the revised form of manuscript. The authors took into consideration all of my remarks.
Reviewer 2 Report
The authors of the paper “Beta-titanium alloy covered by ferroelectric coating – 2 physicochemical properties and human osteoblast-like cell 3 response” tried to bring new scientific evidence to support the conclusions of the first form of the article and thus respond to the reviewers' request.
Thus, the problems related to the existence of a layer of ferroelectric barium titanate with a thickness of one micron and which can also present spontaneous polarization have been clarified.
The benefits of the barium titanate interface layer between the cell culture medium and the titanium alloy coated with titanium dioxide are not important at this stage of the work, but once the working hypotheses are correctly established, there is a chance for substantial results for the future.
Considering all the modifications made to the paper, the decision is to be accepted for publication in Coatings.